# I4VGEN: IMAGE AS FREE STEPPING STONE FOR TEXT-TO-VIDEO GENERATION

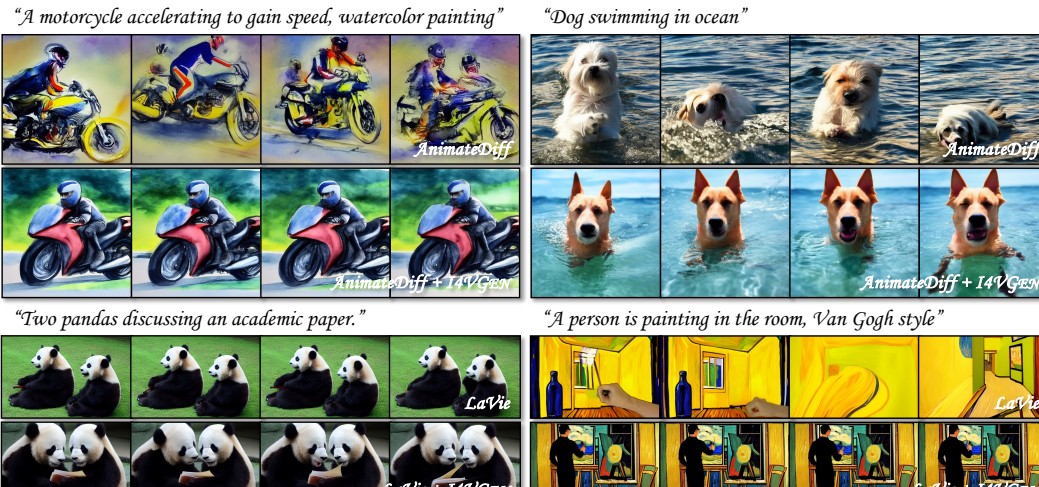

Figure 1: **Example results** synthesized by the proposed I4VGEN. I4VGEN is seamlessly integrated into existing pre-trained text-to-video diffusion models without additional training, significantly improving the temporal consistency (*e.g.*, top-left and bottom-right), visual realism (*e.g.*, top-right), and semantic fidelity (*e.g.*, bottom-left) of the synthesized videos.

## ABSTRACT

Text-to-video generation has trailed behind text-to-image generation in terms of quality and diversity, primarily due to the inherent complexities of spatio-temporal modeling and the limited availability of video-text datasets. Recent text-to-video diffusion models employ the image as an intermediate step, significantly enhancing overall performance but incurring high training costs. In this paper, we present I4VGEN, a novel video diffusion inference pipeline to leverage advanced image techniques to enhance pre-trained text-to-video diffusion models, which requires no additional training. Instead of the vanilla text-to-video inference pipeline, I4VGEN consists of two stages: anchor image synthesis and anchor image-augmented text-to-video synthesis. Correspondingly, a simple yet effective generation-selection strategy is employed to achieve visually-realistic and semantically-faithful anchor image, and an innovative noise-invariant video score distillation sampling (NI-VSDS) is developed to animate the image to a dynamic video by distilling motion knowledge from video diffusion models, followed by a video regeneration process to refine the video. Extensive experiments show that the proposed method produces videos with higher visual realism and textual fidelity. Furthermore, I4VGEN also supports being seamlessly integrated into existing image-to-video diffusion models, thereby improving overall video quality.

## 1 INTRODUCTION

Recent advances in large-scale text-to-image diffusion models (Esser et al., 2021; Balaji et al., 2022; Ramesh et al., 2022; Nichol et al., 2022; Saharia et al., 2022; Feng et al., 2023; Gu et al., 2023; Xue

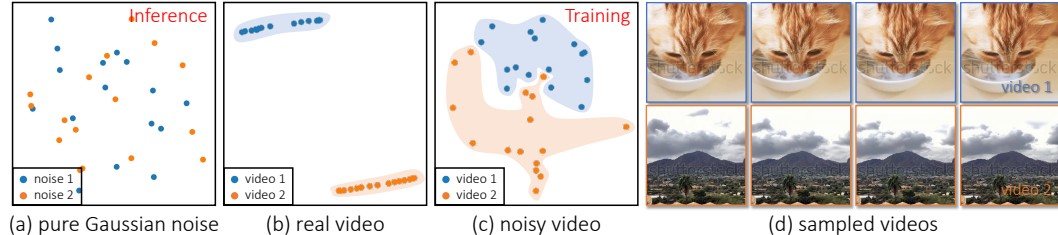

Figure 2: **Illustration of non-zero terminal signal-to-noise ratio.** We employ t-SNE to visualize the distributions of pure Gaussian noise, real video, and noisy video at the timestep $T$, where each data point represents an independently sampled noise point or video frame. The noise schedule of AnimateDiff (Guo et al., 2024b) is used, and all operations are performed in the latent space of the video autoencoder. (a) The distribution of pure Gaussian noise exhibits a disordered and diffuse nature; (b) real videos are temporally-correlated and different videos can be clearly distinguished from each other; (c) noisy videos preserve a certain degree of temporal correlation and maintain separability between different videos; (d) sampled videos for visualization.

et al., 2023) have demonstrated the capability to generate diverse and high-quality images from extensive web-scale image-text pair datasets. Efforts to extend these diffusion models to text-to-video synthesis (Ho et al., 2022a; Zhou et al., 2022; Chen et al., 2023a; Singer et al., 2023; Wang et al., 2023b;e; Blattmann et al., 2023a; Girdhar et al., 2023; Guo et al., 2024b; Bao et al., 2024) have involved leveraging video-text pairs and temporal modeling. However, text-to-video generation remains inferior to image counterpart in terms of both quality and diversity, primarily due to the complex nature of spatio-temporal modeling and the limited size of video-text datasets, which are often an order of magnitude smaller than image-text datasets.

This paper explores a novel video diffusion inference pipeline that leverages advanced image techniques to enhance pre-trained text-to-video diffusion models, focusing on the following two insights:

**Image conditioning for text-to-video generation.** Recent methods (Blattmann et al., 2023a; Zhang et al., 2023b; Girdhar et al., 2023; Chen et al., 2024a; Li et al., 2023; Hu et al., 2023) have adopted image-guided text-to-video generation, where an initial image generation step significantly enhances video output quality. This paradigm benefits from the strong capabilities of text-to-image models by using the generated images as detailed references for video synthesis. While effective, these approaches incurs additional high training costs. This paper builds on this insight but innovates by designing a novel video diffusion inference pipeline to leverage image information, thereby enhancing text-to-video generation performance without additional training expense.

**Zero terminal-SNR noise schedule.** A prevalent issue in diffusion models is the non-zero terminal signal-to-noise ratio (SNR) (Guttenberg; Lin et al., 2024). The mismatch between the training phase, where residual signals persist in noisy videos at the terminal diffusion timestep $T$, and the inference phase, which uses pure Gaussian noise at the timestep $T$, creates a gap that degrade the model performance. As illustrated in Fig. 2, noisy videos exhibit temporal correlation that is distinctly different from the independent and identically distributed pure Gaussian noise. This paper is dedicated to reconfiguring the inference pipeline to circumvent this issue.

Motivated by these insights, we propose a novel video diffusion inference pipeline, called I4VGEN, which enhances pre-trained text-to-video diffusion models by incorporating image information into the inference process. This method requires no additional learnable parameters and training costs, and can be seamlessly integrated into existing text-to-video diffusion models, circumventing the non-zero terminal SNR issue and improving output quality.

Specifically, instead of the vanilla text-to-video inference pipeline, which fails to leverage image reference information, I4VGEN decomposes the inference process into two stages: anchor image synthesis and anchor image-augmented text-to-video synthesis. For the former, a simple yet effective generation-selection strategy is introduced, which involves synthesizing candidate images and selecting the most suitable one using a reward-based mechanism, thereby obtaining a visually-realistic anchor image that is closely aligned with the text prompt. For the latter, we develop an innovative noise-invariant video score distillation sampling (NI-VSDS) to animate the anchor image

to a dynamic video by extracting motion knowledge from text-to-video diffusion models, followed by a video regeneration process, *i.e.*, diffusion-denoising, to refine the video. This inference pipeline avoids the issue of non-zero terminal SNR.

Extensive quantitative and qualitative analyses demonstrate that I4VGEN can be effectively applied to various text-to-video diffusion models, significantly improving the temporal consistency, visual realism, and semantic fidelity of the synthesized videos (see Fig. 1). Moreover, our method can also be seamlessly integrated into existing image-to-video diffusion models, thereby enhancing the temporal consistency and visual quality of the generated videos (see Fig. 6).

The main novelties and contributions are as follows:

- We propose a novel video diffusion inference pipeline, called I4VGEN, which enhances pre-trained text-to-video diffusion models by incorporating image reference information into the inference process, without requiring additional training or learnable parameters.
- We employ a simple yet effective generation-selection strategy to achieve high-quality image, and design a novel noise-invariant video score distillation sampling for image animation.
- We comprehensively evaluate our approach with representative text-to-video diffusion models, and demonstrate I4VGEN significantly improves the quality of generated videos. Furthermore, I4VGEN can also be adapted to image-to-video diffusion models, leading to improved results.

## 2 PRELIMINARIES

**Video diffusion models.** Aligned with the framework of image diffusion models, Video diffusion models (VDMs) predominantly utilize the paradigm of latent diffusion models (LDMs). Unlike traditional methods that operate directly in the pixel space, VDMs function within the latent space defined by a video autoencoder. Specifically, a video encoder $\mathcal{E}(\cdot)$ learns the mapping from an input video $\mathbf{v} \in \mathcal{V}, \mathbf{v} = \{\mathbf{f}^1, \mathbf{f}^2, \cdots, \mathbf{f}^F\}$ to a latent code $\mathbf{z} = \mathcal{E}(\mathbf{v}) = \{\mathbf{z}^1, \mathbf{z}^2, \cdots, \mathbf{z}^f\}$. Subsequently, a video decoder $\mathcal{D}(\cdot)$ reconstructs the input video, aiming for $\mathcal{D}(\mathcal{E}(\mathbf{v})) \approx \mathbf{v}$. Typically, image autoencoder is used in a frame-by-frame processing manner instead of the video one, where $F = f$.

Upon training the autoencoder, a Denoising Diffusion Probabilistic Model (DDPM) (Ho et al., 2020) is employed within the latent space to generate a denoised version of an input latent $\mathbf{z}_t$ at each timestep $t$. During denoising, the diffusion model can be conditioned on additional inputs, such as a text embedding $\mathbf{c} = f_{\text{CLIP}}(\mathbf{y})$ generated by a pre-trained CLIP text encoder (Radford et al., 2021), corresponding to the input text prompt $\mathbf{y}$. The DDPM model $\epsilon_\theta(\cdot)$, a 3D U-Net parametrized by $\theta$, optimizes the following loss:

$$\mathcal{L} = \mathbb{E}_{\mathbf{z} \sim \mathcal{E}(\mathbf{v}), \mathbf{c} = f_{\text{CLIP}}(\mathbf{y}), \epsilon \sim \mathcal{N}(\mathbf{0}, \mathbf{1}), t} \left[ \| \epsilon - \epsilon_\theta(\mathbf{z}_t, \mathbf{c}, t) \|_2^2 \right], \tag{1}$$

During inference, a latent variable $\mathbf{z}_T$ is sampled from the standard Gaussian distribution $\mathcal{N}(\mathbf{0}, \mathbf{1})$ and subjected to sequential denoising procedures of the DDPM to derive a refined latent $\mathbf{z}_0$. This denoised latent $\mathbf{z}_0$ is then fed into the decoder to synthesize the corresponding video $\mathcal{D}(\mathbf{z}_0)$.

**Score distillation sampling.** Score distillation sampling (SDS) (Poole et al., 2023; Wang et al., 2023a) employs the priors of pre-trained text-to-image models to facilitate text-conditioned 3D generation. Specifically, given a pre-trained diffusion model $\epsilon_\theta(\cdot)$ and the conditioning embedding $\mathbf{c} = f_{\text{CLIP}}(\mathbf{y})$ corresponding to the text prompt $\mathbf{y}$, SDS optimizes a set of parameters $\phi$ of a differentiable parametric image generator $\mathcal{G}(\cdot)$ (*e.g.*, NeRF (Mildenhall et al., 2020)) using the gradient of the SDS loss $\mathcal{L}_{\text{SDS}}$:

$$\nabla_\phi \mathcal{L}_{\text{SDS}} = w(t) \left( \epsilon_\theta(\mathbf{z}_t, \mathbf{c}, t) - \epsilon \right) \frac{\partial \mathbf{x}}{\partial \phi}, \tag{2}$$

where $\epsilon$ is sampled from $\mathcal{N}(\mathbf{0}, \mathbf{1})$, $\mathbf{x}$ is an image rendered by $\mathcal{G}$, $\mathbf{z}_t$ is obtained by adding Gaussian noise $\epsilon$ to $\mathbf{x}$ corresponding to the timestep $t$ of the diffusion process, $w(t)$ is a constant that depends on the noising schedule. Inspired by this method, we proposes a noise-invariant video score distillation sampling (NI-VSDS) strategy to efficiently harness the motion prior learned by the text-to-video diffusion model.

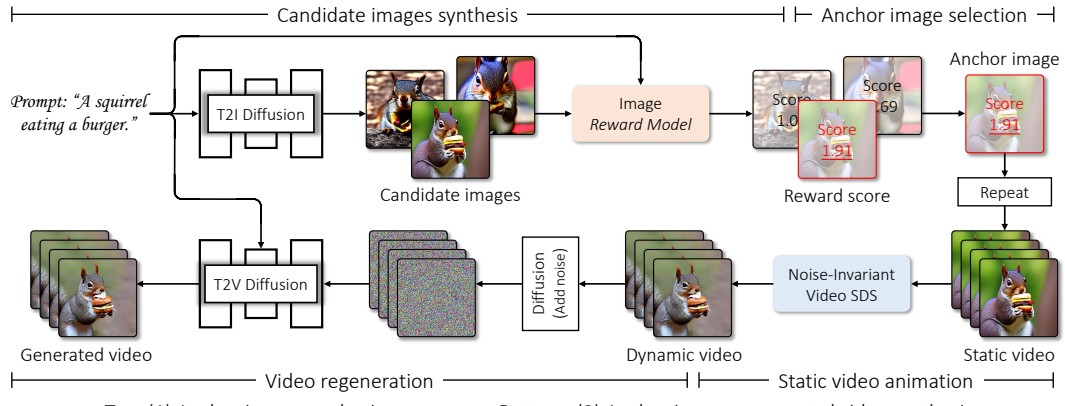

Top: (1) Anchor image synthesis    Bottom: (2) Anchor image-augmented video synthesis

Figure 3: **Illustration of I4VGEN.** I4VGEN is a novel video diffusion inference pipeline, which enhances pre-trained text-to-video diffusion models by incorporating image reference information into the inference process. Instead of the vanilla text-to-video inference pipeline, I4VGEN consists of two stages: (1) anchor image synthesis and (2) anchor image-augmented text-to-video synthesis. Firstly, a simple yet effective generation-selection strategy is applied to synthesize candidate images and select the most suitable image using a reward-based mechanism, thereby obtaining high-quality anchor image. Subsequently, an innovative noise-invariant video scoring distillation sampling (NI-VSDS) is developed, which extracts motion prior from the text-to-video diffusion model to animate the anchor image into dynamic video, followed by a video regeneration process to refine the video.

## 3    I4VGEN

This section introduces **I4VGEN**, a novel video diffusion inference pipeline designed for enhancing the capabilities of pre-trained text-to-video diffusion models. As illustrated in Fig. 3, we factorize the inference process into two stages: (1) anchor image synthesis to generate the anchor image $\mathbf{x}$ given the text prompt $\mathbf{y}$, and (2) anchor image-augmented video synthesis to generate the video $\mathbf{v}$ by leveraging the text prompt $\mathbf{y}$ and the anchor image $\mathbf{x}$. This section provides the detailed explanations of both stages in Sec. 3.1 and 3.2, respectively.

### 3.1    ANCHOR IMAGE SYNTHESIS

The goal of this stage is to synthesize visually-realistic anchor images $\mathbf{x}$ that accurately correspond to the given text prompts $\mathbf{y}$. This image serves as a foundation to provide appearance information for enhancing the performance of the subsequent video generation. As illustrated in Fig. 3 (Top), a simple yet effective generation-selection pipeline is employed to produce the anchor image, which consist of candidate images synthesis and reward-based anchor image selection.

**Candidate images synthesis.** Instead of generating a single image, our approach produces a set of candidate images to ensure the selection of the best example. Utilizing a pre-trained image diffusion model $\mathcal{D}_{\mathrm{img}}(\cdot)$, we construct the candidate image set as follows:

$$\mathbf{x}_1, \mathbf{x}_2, \cdots, \mathbf{x}_N = \mathcal{D}_{\mathrm{img}}\left(\mathbf{y}, \mathbf{z}_1\right), \mathcal{D}_{\mathrm{img}}\left(\mathbf{y}, \mathbf{z}_2\right), \cdots, \mathcal{D}_{\mathrm{img}}\left(\mathbf{y}, \mathbf{z}_N\right), \qquad (3)$$

where $N$ denotes the number of candidate images, and $\mathbf{z}_i$ represents Gaussian noise.

**Reward-based anchor image selection.** With the help of the image reward model $\mathcal{R}(\cdot)$ (Xu et al., 2023), a promising automatic text-to-image evaluation metric aligned with human preferences, the candidate image with the highest reward score $s$ is selected as the anchor image $\mathbf{x}$, as defined by:

$$\mathbf{x} = \mathbf{x}_i, \quad \text{where } i = \arg\max_i s_i = \arg\max_i \mathcal{R}\left(\mathbf{x}_i\right). \qquad (4)$$

The generation-selection design facilitates the acquisition of a high-quality anchor image, particularly beneficial for complex text prompts (see Fig. 5). Notably, our method accommodates both user-provided and retrieved images, extending its applicability to a variety of custom scenarios, as discussed in Sec. 4.5.

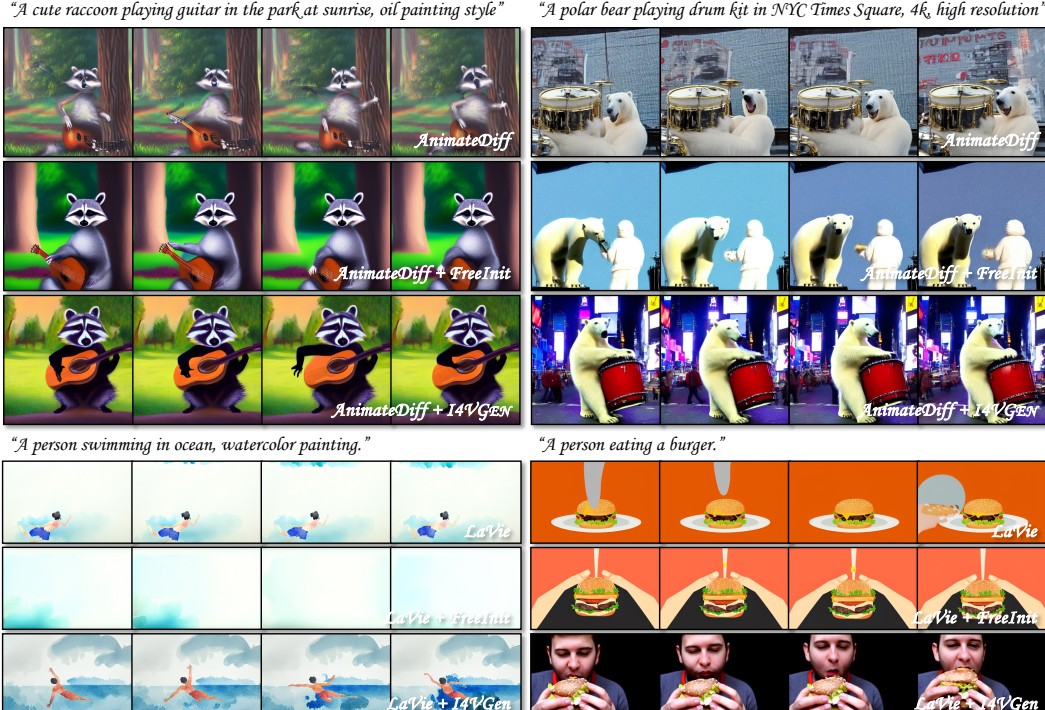

Figure 4: **Qualitative comparison.** Each video is generated with the same text prompt and random seed for all methods. Our approach significantly improves the quality of the generated videos while showing excellent alignment with text prompts.

## 3.2 ANCHOR IMAGE-AUGMENTED VIDEO SYNTHESIS

Upon obtaining the anchor image $\mathbf{x}$, we replicate it $F$ times to create an initial static video $\hat{\mathbf{v}} \in \mathcal{V}, \hat{\mathbf{v}} = \{\mathbf{x}, \mathbf{x}, \cdots, \mathbf{x}\}$. The goal of this stage is to convert this static video into a high-quality video reflecting the text prompt $\mathbf{y}$. As illustrated in Fig. 3 (Bottom), we introduce static video animation and video regeneration.

**Static video animation.** A straightforward approach to animate the static video involves applying a diffusion-denoising process to transition from the static to dynamic state. However, this approach still encounters a training-inference gap, as the text-to-video diffusion model is trained on dynamic real-world videos but tested on static videos, leading to sub-optimal motion quality due to the introduction of static priors, as discussed in Sec. 4.4.

To address this limitation, we propose a novel approach leveraging the motion prior from the pre-trained text-to-video diffusion model to animate static videos. Drawing inspiration from score distillation sampling (SDS) as introduced in (Poole et al., 2023; Wang et al., 2023a), we develop the noise-invariant video score distillation sampling (NI-VSDS). Unlike vanilla SDS, which optimizes a parametric image generator, our approach directly parameterizes the static video $\hat{\mathbf{v}}$ and applies targeted optimization to it. The NI-VSDS loss function is defined as follows:

$$\nabla_{\hat{\mathbf{v}}} \mathcal{L}_{\text{NI-VSDS}} = w(t) \left( \epsilon_\theta(\hat{\mathbf{v}}_t, \mathbf{c}, t) - \epsilon \right), \tag{5}$$

where $\hat{\mathbf{v}}_t$ represents the noisy video at timestep $t$ perturbed by Gaussian noise $\epsilon$. Furthermore, we incorporate three strategic modifications:

- Instead of resampling the Gaussian noise at each iteration as in traditional SDS, we maintain a constant noise across the optimization, enhancing convergence speed.
- Optimization is confined to the initial stages of the denoising process, where noise levels are higher, focusing on dynamic information distillation.
- We implement a coarse-to-fine optimization strategy, evolving from high to low noise levels, specifically from timestep $T$ to $\tau_{\text{NI-VSDS}}$, where $T > \tau_{\text{NI-VSDS}} > 0$. This approach stabilizes the optimization trajectory and yields superior motion quality.

Table 1: **VBench evaluation results per dimension.** This table compares the performance of I4VGEN with other counterparts across each of the 16 VBench dimensions.

| Methods | Subj. Cons. | Back. Cons. | Tem. Flick. | Moti. Smo. | Dyna. Degr. | Aest. Qual. | Imag. Qual. | Obje. Class |
|---|---|---|---|---|---|---|---|---|
| AnimateDiff | 87.11% | 95.22% | 95.99% | 93.12% | **74.89%** | 56.07% | 64.29% | 83.69% |
| + FreeInit | 90.45% | 96.57% | 96.89% | 95.66% | 70.17% | 59.25% | 63.51% | 87.55% |
| + I4VGEN | **95.17%** | **97.73%** | **98.51%** | **96.45%** | 57.72% | **64.68%** | **66.18%** | **92.59%** |
| LaVie | 91.65% | 96.30% | 98.03% | 95.73% | **71.94%** | 59.64% | 65.13% | 91.25% |
| + FreeInit | 92.32% | 96.35% | 98.06% | 95.83% | 71.11% | 59.41% | 63.89% | 89.13% |
| + I4VGEN | **94.12%** | **96.90%** | **98.55%** | **96.37%** | 70.55% | **60.88%** | **66.55%** | **92.26%** |

| Methods | Mult. Obje. | Hum. Acti. | Color | Spat. Rela. | Scene | Appe. Style | Tem. Style | Over. Cons. |
|---|---|---|---|---|---|---|---|---|
| AnimateDiff | 22.61% | 90.40% | 81.73% | 31.55% | 45.61% | 24.40% | 24.49% | 25.71% |
| + FreeInit | 26.92% | 93.00% | 86.39% | 30.71% | 44.61% | 23.98% | 25.03% | 25.61% |
| + I4VGEN | **57.22%** | **95.80%** | **91.98%** | **45.20%** | **54.67%** | **25.07%** | **26.11%** | **28.01%** |
| LaVie | 24.02% | 94.80% | 83.64% | 26.27% | 52.89% | 23.67% | 24.94% | 27.25% |
| + FreeInit | 22.59% | 94.20% | 84.34% | 27.46% | 52.70% | 23.61% | 24.85% | 26.89% |
| + I4VGEN | **32.77%** | **96.20%** | **88.59%** | **33.81%** | **55.64%** | **24.35%** | **25.62%** | 27.68% |

The implementation of noise-invariant video score distillation sampling (NI-VSDS) algorithm is detailed in Algorithm 1, which outlines the process of converting a static video into a dynamic video using the defined NI-VSDS loss. Notably, we only perform a single update from timestep $T$ to $\tau_{\text{NI-VSDS}}$, requiring fewer than 50 iterations, this is a significant reduction compared to the thousands of iterations typically required for text-to-3D synthesis in SDS. $\alpha$ is a scalar that defines the step size of the gradient update. We empirically set $\tau_{\text{NI-VSDS}} = \text{Int}(T \times p_{\text{NI-VSDS}})$.

---

**Algorithm 1: NI-VSDS**

---

**Input:** T2V diffusion model $\epsilon_\theta(\cdot)$, text prompt $\mathbf{y}$, static video $\hat{\mathbf{v}}$, timestep $\tau_{\text{NI-VSDS}}$.

**Output:** Dynamic video.

1   Sampling $\epsilon \sim \mathcal{N}(\mathbf{0}, \mathbf{1})$; $\mathbf{c} = f_{\text{CLIP}}(\mathbf{y})$
2   **for** $t = T, \cdots, \tau_{\text{NI-VSDS}}$ **do**
3      $\hat{\mathbf{v}}_t \leftarrow \text{AddNoise}(\hat{\mathbf{v}}, \epsilon, t)$
4      $\nabla_{\hat{\mathbf{v}}} \mathcal{L}_{\text{NI-VSDS}} \leftarrow w(t)\left(\epsilon_\theta(\hat{\mathbf{v}}_t, \mathbf{c}, t) - \epsilon\right)$
5      $\hat{\mathbf{v}} \leftarrow \hat{\mathbf{v}} - \alpha \cdot \nabla_{\hat{\mathbf{v}}} \mathcal{L}_{\text{NI-VSDS}}$

6   **return** $\hat{\mathbf{v}}$

---

**Video regeneration.** After animating the static video, we further enhance the appearance detail quality of the video through a diffusion-denoising process. This stage is not affected by the aforementioned training-inference gap, thereby achieving more refined generation results.

Notably, we can flexibly add noise up to any timestep $\tau_{\text{re}}$, calculated as $\tau_{\text{re}} = \text{Int}(T \times p_{\text{re}})$, followed by the corresponding denoising process. This strategy not only preserves the fine appearance textures but also reduces the required denoising steps, thus streamlining the video synthesis process and elevating the overall quality of the resulting video.

## 4 EXPERIMENTS

### 4.1 EXPERIMENTAL SETTINGS

**Implementation details.** I4VGEN a novel video diffusion inference pipeline that leverages advanced image techniques to enhance pre-trained text-to-video diffusion models without requiring additional training, and can be seamlessly integrated into existing text-to-video diffusion models. To ascertain the efficacy and adaptability of I4VGEN, we apply it to two well-regarded text-to-video diffusion models: AnimateDiff (Guo et al., 2024b) and LaVie (Wang et al., 2023e).

For AnimateDiff, the `mm-sd-v15-v2` motion module[1], alongside Stable Diffusion v1.5, is utilized to synthesize 16 consecutive frames at a resolution of $512 \times 512$ pixels for evaluation. For LaVie, the

---

[1]https://github.com/guoyww/AnimateDiff

Table 2: **Ablation study.** Orange highlights generation-selection, while yellow highlights NI-VSDS.

| Methods | Subj. Cons. | Back. Cons. | Tem. Flick. | Moti. Smo. | Dyna. Degr. | Aest. Qual. | Imag. Qual. | Obje. Class |
|---|---|---|---|---|---|---|---|---|
| AnimateDiff | 87.11% | 95.22% | 95.99% | 93.12% | **74.89%** | 56.07% | 64.29% | 83.69% |
| + I4VGEN (w/o gen.-sel.) | 94.89% | 97.80% | 98.28% | 96.99% | 55.91% | 62.23% | 64.18% | 90.95% |
| + I4VGEN (w/o NI-VSDS) | **96.47%** | **98.82%** | **98.99%** | **97.56%** | 28.24% | **65.17%** | 65.52% | **92.66%** |
| + I4VGEN | 95.17% | 97.73% | 98.51% | 96.45% | 57.72% | 64.68% | **66.18%** | 92.59% |

| Methods | Mult. Obje. | Hum. Acti. | Color | Spat. Rela. | Scene | Appe. Style | Tem. Style | Over. Cons. |
|---|---|---|---|---|---|---|---|---|
| AnimateDiff | 22.61% | 90.40% | 81.73% | 31.55% | 45.61% | 24.40% | 24.49% | 25.71% |
| + I4VGEN (w/o gen.-sel.) | 40.68% | 94.40% | 90.55% | 37.79% | 53.72% | 24.76% | 26.03% | 26.62% |
| + I4VGEN (w/o NI-VSDS) | **62.84%** | 94.80% | 91.95% | **47.57%** | 55.80% | 24.88% | 25.72% | 27.91% |
| + I4VGEN | 57.22% | **95.80%** | **91.98%** | 45.20% | 54.67% | **25.07%** | **26.11%** | **28.01%** |

`base-version`[2] is employed to generate 16 consecutive frames at $320 \times 512$ pixels for evaluation. All other inference details adhere to the original settings described in Guo et al. (2024b) and Wang et al. (2023e), respectively. Notably, both AnimateDiff and LaVie possess inherent text-to-image generation capabilities when excluding the motion module. To avoid introducing additional GPU storage requirements, we leverage their corresponding image versions for text-to-image generation in I4VGEN. For AnimteDiff, we empirically set $N = 16$, $p_{\text{NI-VSDS}} = 0.4$, $\alpha = 1$, and $p_{\text{re}} = 1$. For LaVie, we empirically set $N = 16$, $p_{\text{NI-VSDS}} = 0.4$, $\alpha = 1$, and $p_{\text{re}} = 0.8$. All experiments are conducted on a single NVIDIA V100 GPU (32 GB).

**Benchmark.** I4VGEN is assessed using VBench (Huang et al., 2024), a comprehensive benchmark that evaluates video generation models across 16 disentangled dimensions, which is more authoritative than FVD. These dimensions provide a detailed analysis of generation quality from two overarching perspectives: video quality[3], focusing on the perceptual quality of the generated videos, and video-condition consistency[4], assessing how well the generated videos align with the provided conditions.

## 4.2 Qualitative comparison

Fig. 4 presents a comparative analysis of our results against state-of-the-art counterparts using identical text prompts and random seeds. I4VGEN excels in enhancing both the temporal consistency and the frame-wise quality, alongside superior alignment with the text prompts. For instance, in the case of "playing guitar", AnimateDiff suffers from poor video quality, and FreeInit encounters an incomplete guitar in the middle of the video. In contrast, our method effectively addresses these issues, maintaining stable temporal consistency. Furthermore, while baseline methods struggle with accurate synthesis of all text-described components, *e.g.*, "NYC Times Square", I4VGEN generates videos that are visually realistic and closely aligned with the text prompts by utilizing anchor images obtained by the generation-selection strategy.

## 4.3 Quantitative comparison

**Objective evaluation.** Following the protocols established by VBench, we evaluate I4VGEN in terms of both video quality and video-text consistency. As detailed in Table 1, I4VGEN outperforms all other approaches in temporal quality (higher background and subject consistency, less flickering, and better smoothness), frame-wise quality (higher aesthetic and imaging quality), and video-text

---

[2]`https://github.com/Vchitect/LaVie`

[3]Video quality includes 7 evaluation dimensions: Subject Consistency, Background Consistency, Temporal Flickering, Motion Smoothness, Dynamic Degree, Aesthetic Quality, and Imaging Quality. The first 5 evaluate temporal quality, and the last 2 evaluate frame-wise quality.

[4]Video-condition consistency includes 9 evaluation dimensions: Object Class, Multiple Objects, Human Action, Color, Spatial Relationship, Scene, Appearance Style, Temporal Style, Overall Consistency. The first 6 evaluate semantics, the 7 and 8-th evaluate style, and the 9-th evaluates overall consistency.

*Prompt: "A drone view of celebration with Christmas tree and fireworks, starry sky background"*

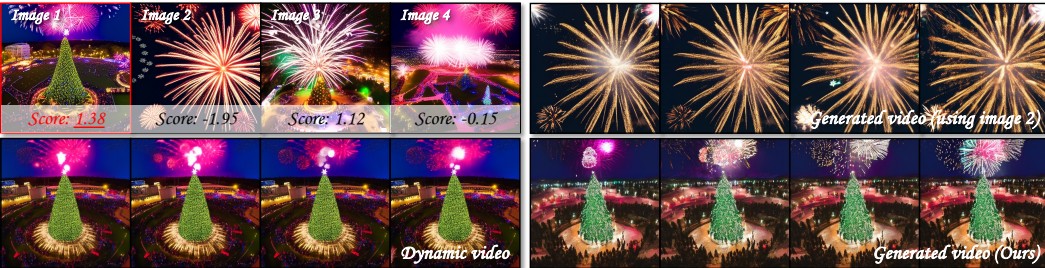

Figure 5: **Intermediate results visualization.** We provide visualizations of the candidate images with reward scores, the dynamic video, and the corresponding generated video.

consistency (greater semantics, style, and overall consistency). Although counterparts occasionally produce videos with more dynamic motion, they are often linked to inappropriate or excessive movements. I4VGEN strikes a more effective balance between motion intensity and overall video quality, which is further verified in the user study.

**User study.** We conduct a subjective user study involving 20 volunteers with expertise in image and video processing, with each participant answering 15 questions. Specifically, participants are asked to select the video with the highest quality across three dimensions: video quality, video-condition consistency, and overall score. As shown in Table 3, our approach outperforms the other methods favorably.

Table 3: **User study.**

| Method | Video Quality | Vid.-Cond. Consistency | Overall score |
|---|---|---|---|
| AnimateDiff | 6.00% | 10.67% | 6.33% |
| + FreeInit | 27.67% | 15.67% | 25.00% |
| + I4VGEN | **66.33%** | **73.67%** | **68.67%** |
| LaVie | 27.67% | 21.33% | 22.33% |
| + FreeInit | 22.67% | 18.33% | 19.67% |
| + I4VGEN | **49.67%** | **60.33%** | **58.00%** |

**Inference time.** We define the time cost of a single denoising iteration for a video in a video diffusion model as $c$. For AnimateDiff (Guo et al., 2024b), following the original inference setting, the time cost to generate a single 16-frame video is $25c$. FreeInit requires 5 rounds of diffusion-denoising to generate a single video, taking a time of $5 \times 25c = 125c$. The time cost for I4VGEN to generate a single video is: $< 25c$ (for synthesizing 16 candidate im-

Table 4: **Inference time.**

| Method | Time |
|---|---|
| AnimateDiff | 21.73s |
| AnimateDiff + FreeInit | 113.67s |
| AnimateDiff + I4VGEN | 53.78s |

ages) $+ 0.6 \times 25c$ (for NI-VSDS) $+ \leq 25c$ (for video regeneration) $= < 65c$ (**total cost**), making it more efficient compared to FreeInit. LaVie (Wang et al., 2023e) shares the same conclusion.

We also provide the inference time for a single video in Table 4, evaluated on a single NVIDIA V100 GPU (32 GB), where 50 videos are randomly generated to obtain an average inference time. Our method performs better than FreeInit.

### 4.4 ABLATION STUDY

**On generation-selection strategy.** We adopt a generation-selection strategy to create visually-realistic and semantically-faithful anchor images, which serve as a foundation for providing appearance information to enhance subsequent video generation performance. As shown in Table 2, highlighted in orange, compared to randomly synthesizing a single anchor image, the generation-selection strategy significantly improves the quality of the generated videos in terms of frame-wise quality and consistency with the text. Fig. 5 provides a visualization of the candidate images, where the reward-based selection strategy eliminates unsatisfactory images, leading to better results.

**On NI-VSDS.** Directly applying the video regeneration process to static videos introduces static priors, resulting in suboptimal motion quality. As shown in Table 2, highlighted in yellow, while direct diffusion-denoising improves the temporal consistency of the generated videos, it severely sacrifices the motion dynamics, adversely affecting the motion style. In contrast, our method achieves an effective balance between motion intensity and overall video quality.

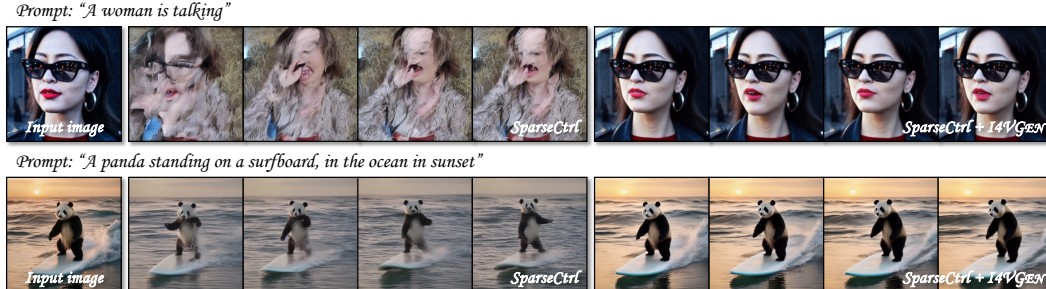

*Prompt: "A woman is talking"*

*Prompt: "A panda standing on a surfboard, in the ocean in sunset"*

Figure 6: **Adaptation on SparseCtrl.** I4VGEN can be seamlessly integrated into SparseCtrl by replacing the anchor image with the provided image, leading to improved results.

**On video regeneration.** Fig. 5 visualizes the intermediate results, demonstrating that the video regeneration process is essential for refining appearance details.

### 4.5 MORE APPLICATIONS

**Adaptation on real image.** Our method adapts to user-provided images, as shown in Fig. 7, where we use real images as anchor images, resulting in high-fidelity videos that are semantically consistent with the real images. Notably, our approach differs from vanilla image-to-video generation, as the synthesized videos are not completely aligned with the provided images. NI-VSDS is designed to animate static videos and is implemented as a spatio-temporal co-optimization.

*Prompt: "A panda drinking coffee in a cafe in Paris"*

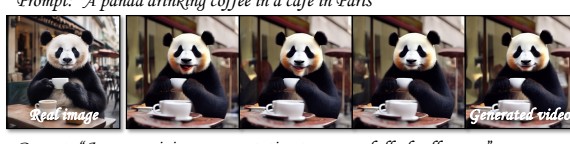

*Prompt: "A person giving a presentation to a room full of colleagues"*

Figure 7: **Adaptation on real image.**

**Adaptation on image-to-video diffusion models.** I4VGEN can be seamlessly integrated into existing image-to-video diffusion models by replacing the anchor images with the provided images, thereby enhancing the overall video quality. As shown in Fig. 6, integrating I4VGEN into SparseCtrl (Guo et al., 2023) significantly improves the quality of the generated videos in terms of temporal consistency and appearance fidelity.

## 5 CONCLUSION

The paper introduces I4VGEN, a novel video diffusion inference pipeline to leverage advanced image techniques to enhance pre-trained text-to-video diffusion models, which requires no additional learnable parameters and training costs. I4VGEN decomposes the text-to-video inference process into anchor image synthesis and anchor image-augmented video synthesis. Correspondingly, a simple yet effective generation-selection strategy is applied to produce a high-quality anchor image, and an innovative noise-invariant video score distillation sampling (NI-VSDS) is designed to animate the image, followed by a video regeneration process to enhance the final output. I4VGEN effectively alleviates non-zero terminal signal-to-noise ratio issues and demonstrates improved visual realism and textual fidelity when integrated with existing video diffusion models.

**Limitation and discussion.** I4VGEN improves the video diffusion model but requires more inference cost. As discussed in Sec. 4.3, the inference time of I4VGEN is over double the baseline. Enhancing inference efficiency remains a future goal, with distillation techniques as a potential approach. Furthermore, removing the generation-selection strategy can reduce inference costs to some extent. As shown in Table 2, our method still significantly outperforms the baseline under this setting. Additionally, although our method and FreeInit are orthogonal, integrating both by replacing video regeneration with FreeInit fails to produce notable benefits.

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

## A    RELATED WORK

**Video Generative Models.** The domain of video generation has seen significant advancements through the use of Generative Adversarial Networks (GANs) (Vondrick et al., 2016; Saito et al., 2017; Tulyakov et al., 2018; Wang et al., 2020; Saito et al., 2020; Tian et al., 2021; Fox et al., 2021; Yu et al., 2022; Skorokhodov et al., 2022; Brooks et al., 2022; Shen et al., 2023; Wang et al., 2023f), Variational Autoencoders (VAEs) (Mittal et al., 2017; Li et al., 2018; He et al., 2018), and Autoregressive models (ARs) (Yan et al., 2021; Ge et al., 2022; Wu et al., 2022; Hong et al., 2023; Villegas et al., 2023; Fu et al., 2023; Yoo et al., 2023; Yu et al., 2023b). Despite these developments, synthesizing videos from text prompts remains challenging due to the complexities of modeling spatio-temporal dynamics. Recent innovations driven by the successes of diffusion models (Ho et al., 2020; Dhariwal & Nichol, 2021; Song et al., 2021), which have been applied effectively in image generation (Rombach et al., 2022; Nichol et al., 2022; Ramesh et al., 2022; Saharia et al., 2022; Gu et al., 2023; Balaji et al., 2022; Xue et al., 2023; Meng et al., 2022; Guo et al., 2024a) and audio synthesis (Kong et al., 2021; Chen et al., 2021; Popov et al., 2021; Leng et al., 2022; Liu et al., 2022), and underscore the emergence of substantial headway (Ho et al., 2022b;a; He et al., 2022; Singer et al., 2023; Blattmann et al., 2023b; Yu et al., 2023c; Ruan et al., 2023; Wu et al., 2023a; Chen et al., 2023a;b; Esser et al., 2023; Ge et al., 2023; Chen et al., 2024a; Geyer et al., 2024; Ma et al., 2023; Wang et al., 2023e; Zhang et al., 2023a;b; Hu et al., 2023; Wang et al., 2023d; Feng et al., 2023; Guo et al., 2024b; Girdhar et al., 2023; Blattmann et al., 2023a; Gupta et al., 2023; Wang et al., 2023b;c; Luo et al., 2023) in research endeavors devoted to video synthesis from text input.

The foundational contributions of the Video Diffusion Model (VDM) (Ho et al., 2022b) represents a milestone in leveraging diffusion models for video generation by adapting the 2D U-Net architecture used in image generation to a 3D U-Net capable of temporal modeling. Successive researches, such as Make-A-Video (Singer et al., 2023) and Imagen Video (Ho et al., 2022a), expand video generation capabilities significantly. To enhance efficiency, subsequent models have transitioned the diffusion process from pixel to latent space (He et al., 2022; Zhou et al., 2022; Wang et al., 2023b; Blattmann et al., 2023b;a; Guo et al., 2024b; Wang et al., 2023e), paralleling advancements in latent diffusion for images (Rombach et al., 2022).

However, the direct generation of videos from text prompts remains intrinsically challenging. Recent approaches (Blattmann et al., 2023a; Zhang et al., 2023b; Girdhar et al., 2023; Chen et al., 2023a; 2024a; Li et al., 2023; Hu et al., 2023; Yu et al., 2023a; Ren et al., 2024) have employed text-to-image synthesis as an intermediary step, enhancing overall performance. Despite these advancements, these methods still face the challenge of high computational training costs. In this study, we explore a novel training-free methodology aimed at bridging the existing gap in the field.

In addition, (Chen et al., 2024b) (contemporary researches) introduces additional operations in the attention layer, *i.e.*, cross-frame self-attention control, to enhance the video model. However, this necessitates modifications to the model architecture, whereas our method does not.

**Signal-Leak Bias.** Diffusion models are designed to generate high-quality visuals from noise through a sequential denoising process, which is consistent in both image and video diffusion models. During training, Gaussian noise corrupts the visual content, challenging the model to restore it to its original form. In the inference phase, the model operates on pure Gaussian noise, transforming it into a realistic visual content step-by-step.

Unfortunately, most existing diffusion models exhibit a disparity between the corrupted image during training and the pure Gaussian noise during inference. Commencing denoising from pure Gaussian noise in the inference phase deviates from the training process, potentially introducing *signal-leak bias*. For image diffusion models, (Guttenberg; Lin et al., 2024; Li et al., 2024) point out flaws in common diffusion noise schedules and sample steps, and propose to fine-tune the diffusion model

to mitigate or eliminate the signal-leak bias during training, leading to improved results. (Everaert et al., 2024) attempts to exploit signal-leak bias to achieve more control over the generated images. For video diffusion models, this issue becomes more pronounced. (Wu et al., 2023b; Ma et al., 2023) invert the retrieved video or generated low-quality to construct initial noise to alleviate the problem of signal-leak, improving inference quality. However, they suffer from limited diversity and cumbersome inference. At the same time, first-round inference of FreeInit (Wu et al., 2023b) still exhibits a training-inference gap.

In contrast to existing methods, our approach utilizes images as the stepping stone for text-to-video generation. This novel pathway aims to produce visually-realistic and semantically-reasonable videos while maintaining manageable computational overheads, as detailed in Sec. 4.

## B  EXPERIMENTS

### B.1  QUALITATIVE COMPARISON

We provide more visualization results in Fig. 8, it can be seen that our method generates more semantically plausible and photo-realistic results than its counterparts. We provide the videos shown in the main paper and appendix in `mp4` format in the Supplementary material.

### B.2  QUANTITATIVE COMPARISON

**On hyperparameters.** I4VGEN is a training-free method that improves video generation performance by correcting the inference process. It is obvious that I4VGEN is also a case-wise method, where different cases correspond to different optimal hyperparameters. In this paper, we provide an empirical setting that is mild for most instances, serving as a performance lower bound for I4VGEN, and facilitating large-scale quantitative comparisons. Furthermore, we also provide a visualization of the impact of hyperparameters in Fig. 9, which shows that carefully tuned hyperparameters can achieve higher-quality videos.

### B.3  FAILURE CASES AND DISCUSSIONS

We provide the failure cases in Fig. 10, I4VGEN is designed to fully unleash the potential of existing video diffusion models, but it still fails to synthesize high-quality videos that are out of the distribution.

## C  CODE

We also provide the code for I4VGEN in the Supplementary material.

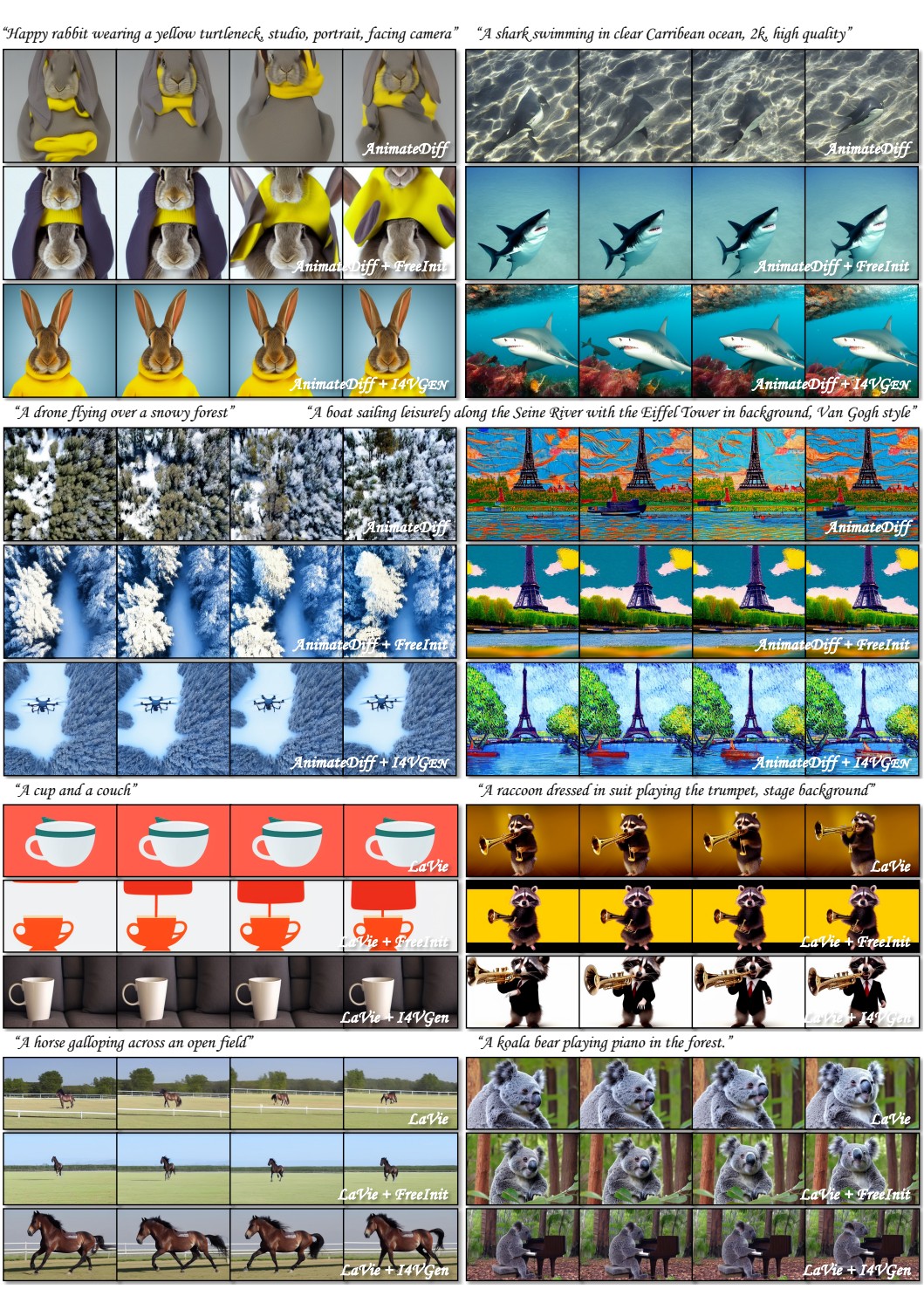

Figure 8: **Qualitative comparison.** Each video is generated with the same text prompt and random seed for all methods. Our approach significantly improves the quality of the generated videos while showing excellent alignment with text prompts.

"A shark swimming in clear Carribean ocean, 2k, high quality"    "A squirrel eating a burger, high quality"

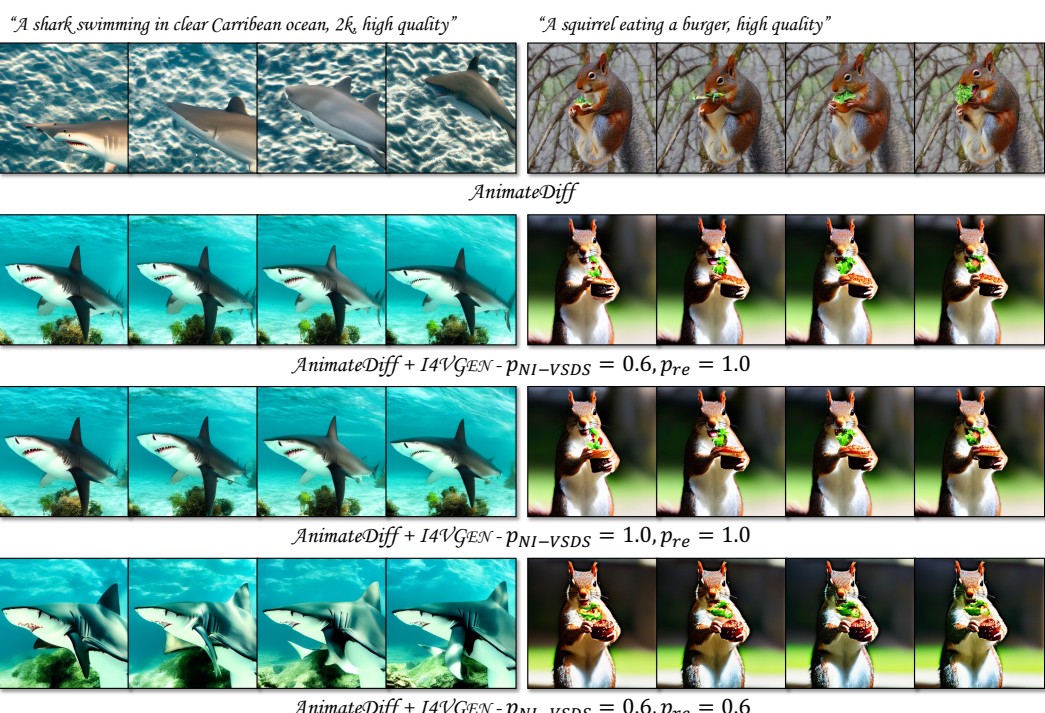

*AnimateDiff*

*AnimateDiff + I4VGEN - $p_{NI-VSDS} = 0.6, p_{re} = 1.0$*

*AnimateDiff + I4VGEN - $p_{NI-VSDS} = 1.0, p_{re} = 1.0$*

*AnimateDiff + I4VGEN - $p_{NI-VSDS} = 0.6, p_{re} = 0.6$*

Figure 9: **Impact of hyperparameters.** For different texts, the optimal parameter settings are different, and the sensitivity to parameters also varies. However, they all significantly outperform the baseline. In this paper, we provide an empirical setting that is mild for most cases, serving as a performance lower bound for I4VGEN. I4VGEN supports fine-tuning parameters on a per-example, achieving higher-quality videos.

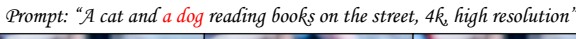

Prompt: "A cat and a dog reading books on the street, 4k, high resolution"

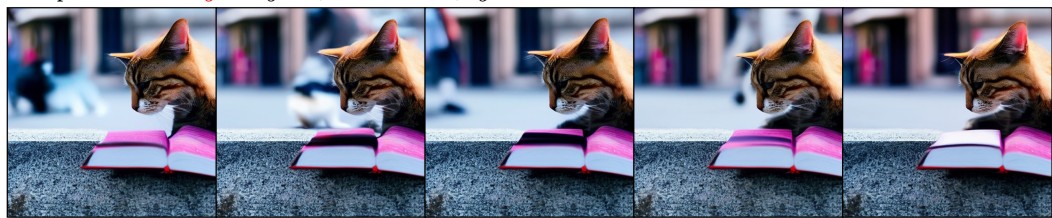

Prompt: "A red cup and a white sofa"

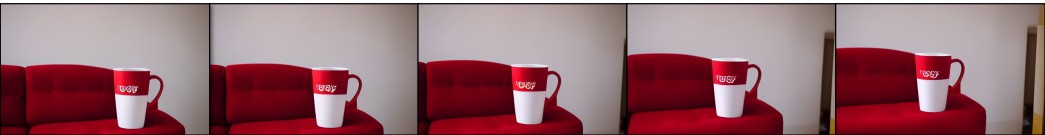

Figure 10: **Failure cases.** I4VGEN is designed to fully unleash the potential of existing video diffusion models, but it still cannot synthesize high-quality videos that are out of the distribution. For example, the text marked in red.

# D  ADDITIONAL EXPERIMENTAL RESULTS

## D.1  ADDITIONAL VISUAL RESULTS ON DYNAMICRAFTER

We integrate I4VGEN into DynamiCrafter (Xing et al., 2024), which exhibits state-of-the-art performance on the VBench Image-to-Video Leaderboard. As shown in Fig. 11, the beginnings and endings of videos generated by DynamiCrafter suffer from low quality. For example, the front part of the face video generated by DynamiCrafter exhibits serious artifacts, and the face in the latter part are deformed. Our method alleviates these issues, which demonstrates that I4VGEN can significantly improve the quality of videos synthesized by DynamiCrafter.

## D.2  ADDITIONAL VISUAL RESULTS ON SPARSECTRL

We conduct experiments on action instructions. As shown in Fig. 12, we explore two prompt-based motion enhancement strategies:

- By providing static descriptions in negative prompt, the dynamic intensity of the synthesized videos can be further enhanced.
- By providing specific action instruction in the prompt, such as "waving its hands", the synthesized video accurately renders this action.

These findings indicate that I4VGEN does not compromise the dynamic nature of the synthesized videos but rather depicts more reasonable and accurate motion.

## D.3  ADDITIONAL VISUAL RESULTS USING FLUX

We provide the visual results of I4VGEN adapted on FLUX in the Fig. 13. Despite the detailed and realistic images synthesized by FLUX, AnimateDiff + I4VGEN is still constrained by the video baseline, *i.e.*, AnimateDiff, in rendering image details and is unable to synthesize realistic videos. Evidently, the distribution of images synthesized by FLUX exceeds what AnimateDiff can handle, which relies on SD 1.5. However, the layout and composition information of images synthesized by FLUX still provide strong support for video synthesis, resulting in promising outcomes.

## D.4  ADDITIONAL VISUAL RESULTS ON ANIMATEDIFF

We provide the visual results on AnimateDiff using the `Realistic Vision V5.1` LoRA in the Fig. 14. Our method still significantly improves the quality of the generated videos while showing excellent temporal consistency.

## D.5  ADDITIONAL INTERMEDIATE RESULTS VISUALIZATION

We provide additional intermediate results in the Fig. 17.

## D.6  ADDITIONAL VISUAL RESULTS ON LARGE CAMERA POSE CHANGE

We provide more visual results involving significant changes in camera poses in Fig. 16, which demonstrate that our method can handle this scenario, improving the temporal consistency and smoothness of the synthesized videos.

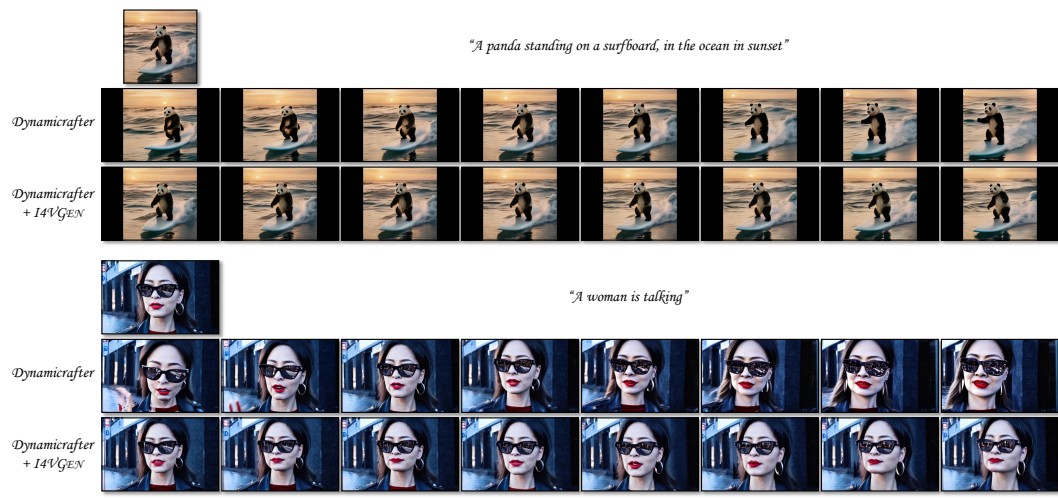

Figure 11: **Additional visual results on DynamiCrafter.** We provide the videos in mp4 format in the supplementary material for better viewing.

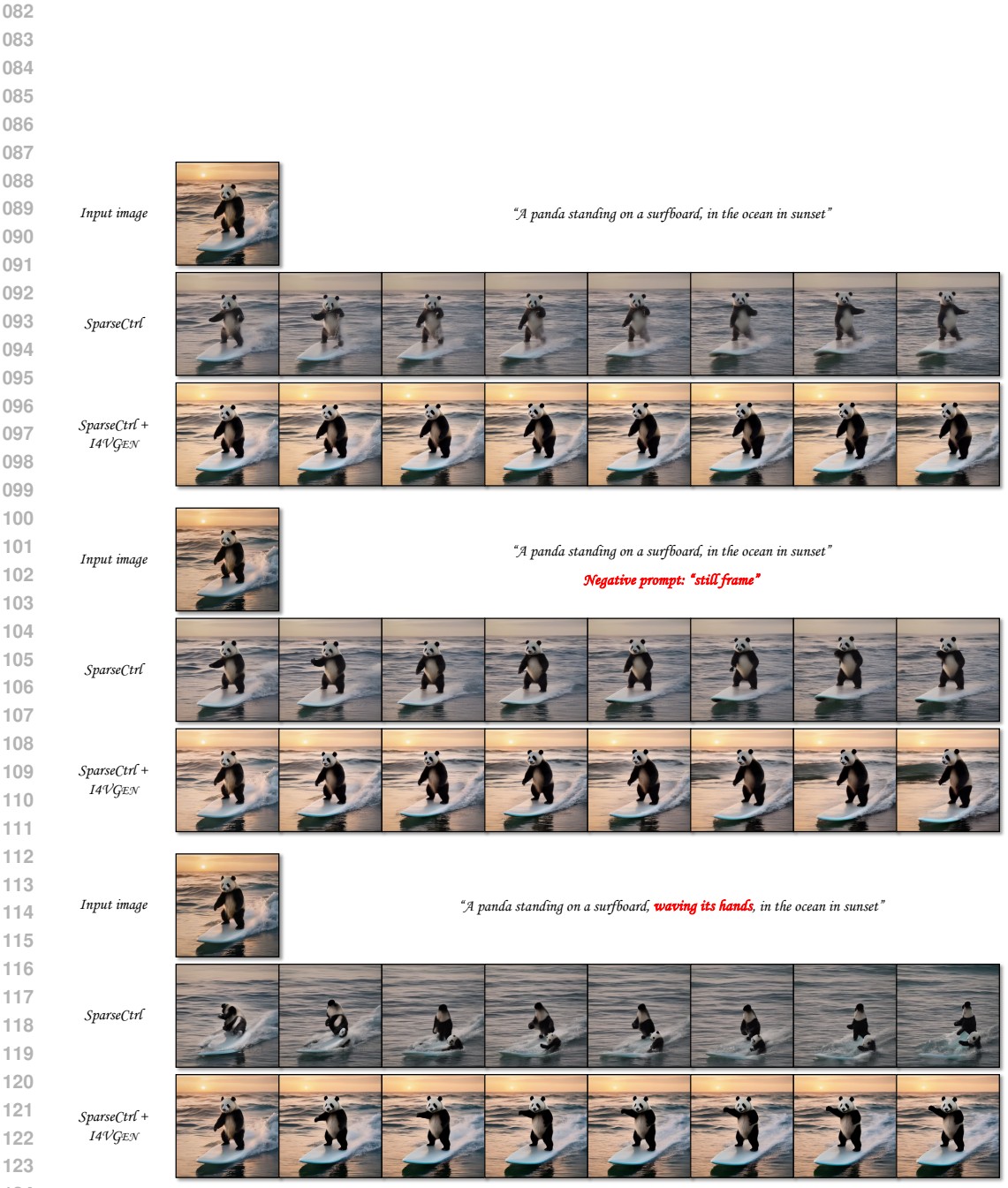

Figure 12: **Additional visual results on SparseCtrl.** We provide the videos in `mp4` format in the supplementary material for better viewing.

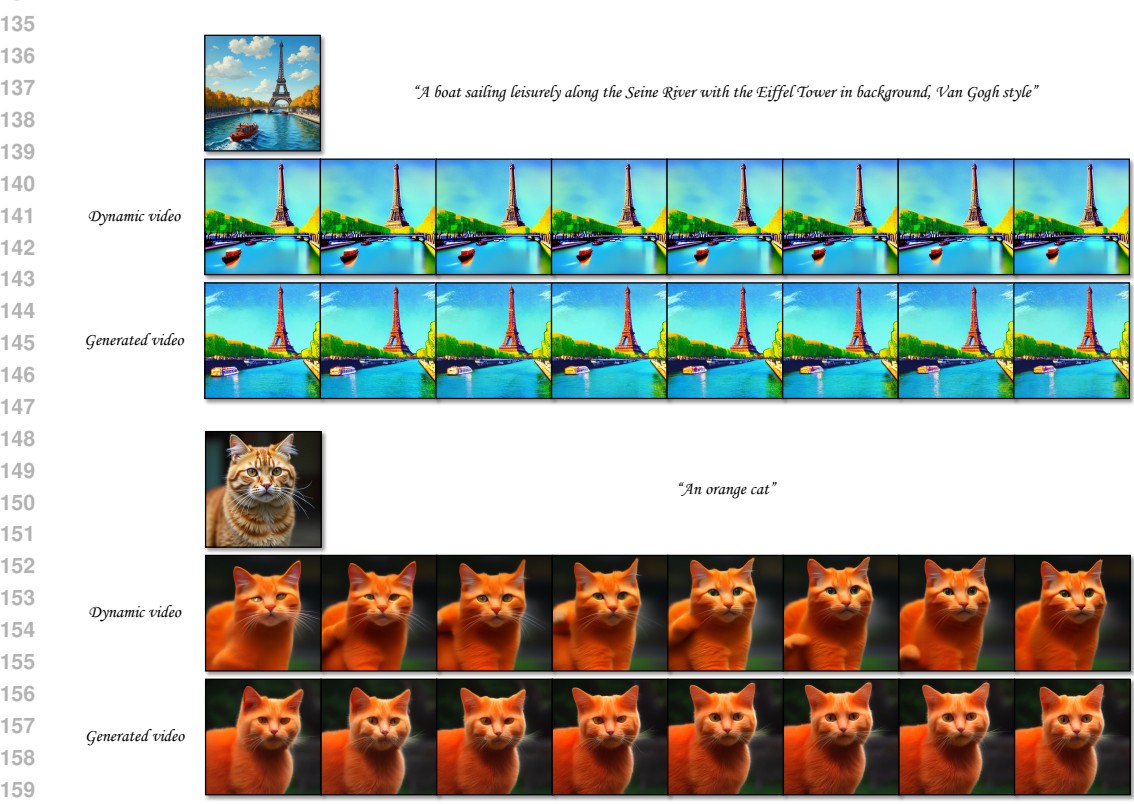

Figure 13: **Adaptation on image generated by FLUX.** We provide the videos in `mp4` format in the supplementary material for better viewing.

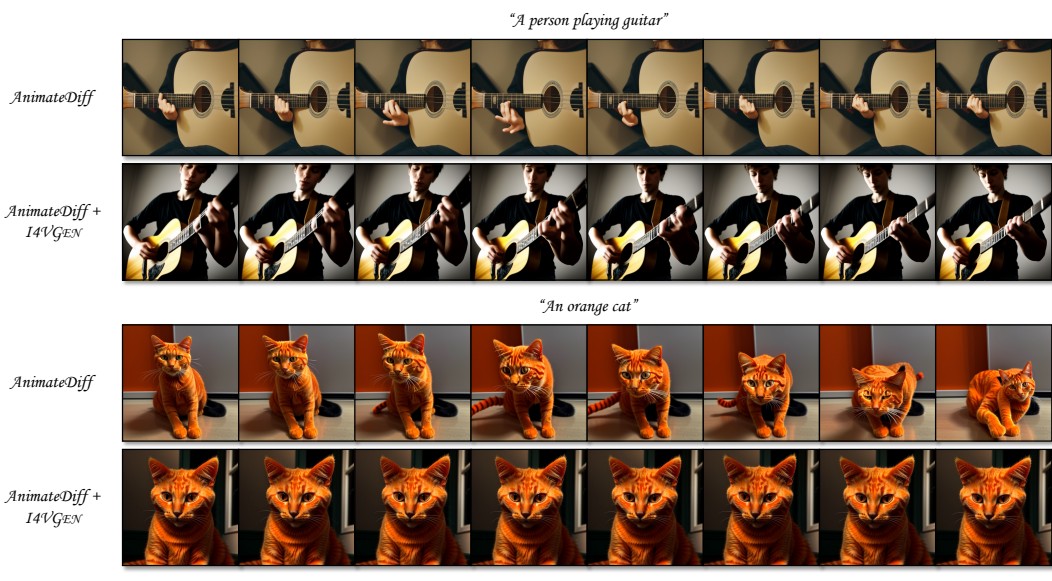

Figure 14: **Additional visual results on AnimateDiff using LoRA.** We provide the videos in `mp4` format in the supplementary material for better viewing.

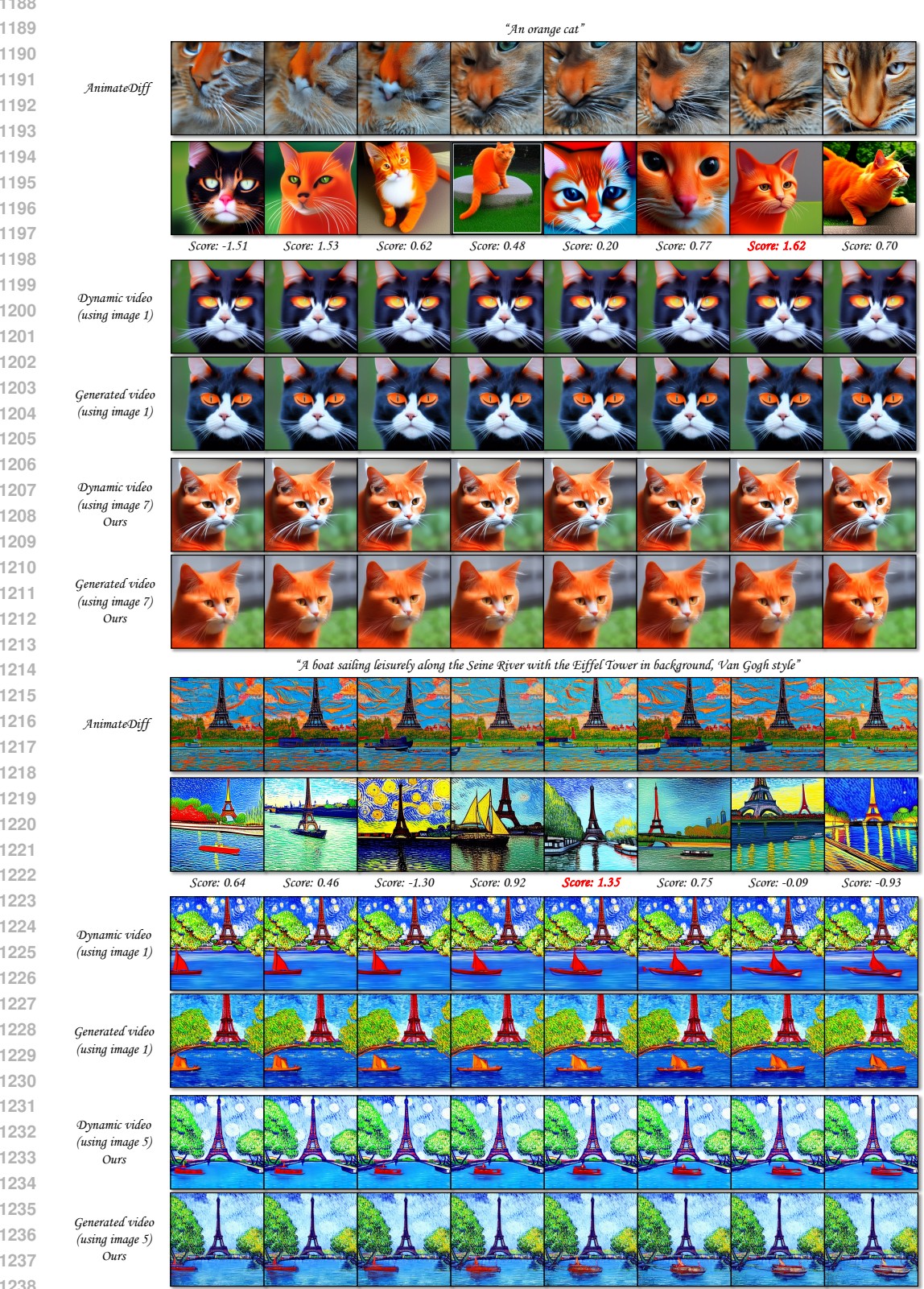

Figure 15: **Additional intermediate results visualization.** We provide the videos in `mp4` format in the supplementary material for better viewing.

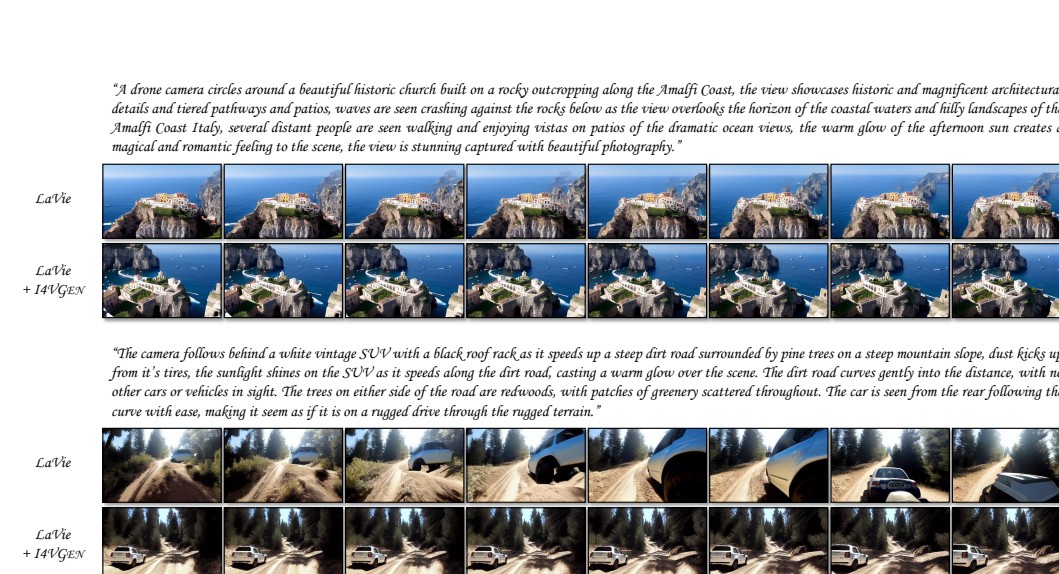

*"A drone camera circles around a beautiful historic church built on a rocky outcropping along the Amalfi Coast, the view showcases historic and magnificent architectural details and tiered pathways and patios, waves are seen crashing against the rocks below as the view overlooks the horizon of the coastal waters and hilly landscapes of the Amalfi Coast Italy, several distant people are seen walking and enjoying vistas on patios of the dramatic ocean views, the warm glow of the afternoon sun creates a magical and romantic feeling to the scene, the view is stunning captured with beautiful photography."*

*"The camera follows behind a white vintage SUV with a black roof rack as it speeds up a steep dirt road surrounded by pine trees on a steep mountain slope, dust kicks up from it's tires, the sunlight shines on the SUV as it speeds along the dirt road, casting a warm glow over the scene. The dirt road curves gently into the distance, with no other cars or vehicles in sight. The trees on either side of the road are redwoods, with patches of greenery scattered throughout. The car is seen from the rear following the curve with ease, making it seem as if it is on a rugged drive through the rugged terrain."*

Figure 16: **Additional visual results on large camera pose change.** We provide the videos in `mp4` format in the supplementary material for better viewing.

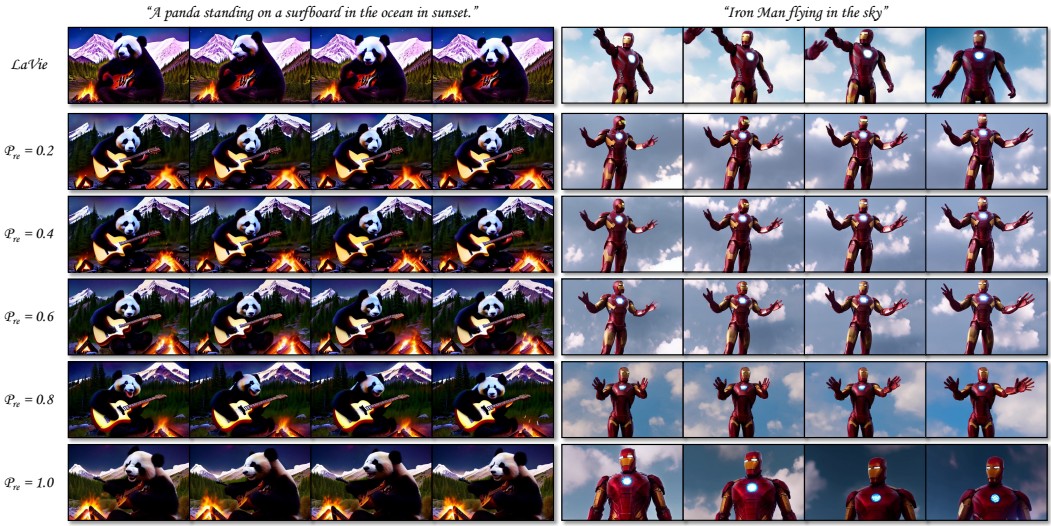

Figure 17: **Impact of $p_{re}$.** We provide the videos in `mp4` format in the supplementary material for better viewing.

