# OpenReview forum: "I4VGen: Image as Free Stepping Stone for Text-to-Video Generation"
_ICLR.cc/2025/Conference — Submitted to ICLR 2025_

### Official Review · Reviewer_tiD5 · 2024-10-30

**Soundness:** 3
**Presentation:** 3
**Contribution:** 3
**Rating:** 6
**Confidence:** 5

**Summary:**

The paper proposes a novel method to initialize the noise for T2V. To endow the noise with motion prior, the paper take the idea of SDS in T-to-3D in a video generation way, which is considered to be novel and inspiring to the community.

**Strengths:**

The paper focuses on the video generation quality from a noise initialization viewpoint, which is hot and also vital to AI-generated content. The authors try to use the off-the-shelf T2I and T2V models in a novel way, to be specific, anchor image synthesis using T2I and motion prior using T2V in a novel SDS way.

**Weaknesses:**

Generally speaking, video generation is notoriously for its lengthy inference cost. Noise initialization adds more cost, which is not practical. The authors are thus encouraged to add more discussion. Besides, the experiments can be more consolidated to enhance the authors' claim. Please refer to the question part for details.

**Questions:**

I have several concerns as follow
1. More proof of the anchor image synthesis. Generally speaking, anchor image act as a low frequency component to mitigate the information leak. A question then rise that whether the anchor image synthesis is necessary. In ablation study, the authors use an example in Fig.5 to validate the assumption. Meanwhile, the authors also confirms that without the generation-selection strategy, the proposed method still performs well. Can the authors give more examples to support this claim?
2. In User study, the 20 volunteers with expertise in image and video processing participated. Does the expert bias exist? that is to say, the participants are more tolerant to the defects of the methods? Will the ordinary participants from a consumer's perspective be better?
3. It's a trend that video generation tends to produce more diverse content with long duration, for example, the large camera pose change. Will the proposed method still work?
4. The proposed method further increase the inference cost. More comprehensive measures are better presented from a practical perspective.

**Details Of Ethics Concerns:**

The authors are encouraged to append the potentially malicious application in generating fake content

---

> ### Author Response · Authors · 2024-11-23
> **Thank you for your valuable feedbacks.**
>
> We would like to thank Reviewer for the constructive review. We did our best within the time constraints to address all of the points that you have raised, and will do our best to answer other concerns.
>
> > **1. Inference cost.**
>
> As discussed in the "Limitation and Discussion" section, I4VGen enhances the video diffusion model but comes with increased inference costs. Simultaneously, we also present a relatively low-cost configuration by removing the generation-selection strategy, i.e., AnimateDiff + I4VGen (w/o gen.-sel.) in Table 2, which still significantly outperforms the baseline. The inference times for this configuration are provided in the table below:
>
> | Method | Inference time |
> | :--- | :---: |
> | AnimateDiff | 21.73s |
> | AnimateDiff + I4VGen (w/o gen.-sel.) | 38.39s |
> | AnimateDiff + I4VGen | 53.78s |
>
> Compared to the full I4VGen, I4VGen (w/o gen.-sel.) offers a more economical option, providing users with additional trade-off choices.
>
> > **2. More proof of the anchor image synthesis. Generally speaking, anchor image act as a low frequency component to mitigate the information leak. A question then rise that whether the anchor image synthesis is necessary. In ablation study, the authors use an example in Fig.5 to validate the assumption. Meanwhile, the authors also confirms that without the generation-selection strategy, the proposed method still performs well. Can the authors give more examples to support this claim?**
>
> Thank you for the reviewers' suggestions. We have provided additional examples in the **Fig. 15 of the supplementary material**.
>
> For the example "An orange cat", if we mistakenly synthesize an image of a black cat using text-to-image diffusion model for subsequent video synthesis, it would greatly lead to the video incorrectly depicting a black cat. Therefore, it is crucial to use the generation-selection strategy to obtain the higher quality and fidelity anchor image.
>
> Moreover, when the generation-selection strategy is removed, as image models outperform video models, text-to-image still has a higher probability of accurately synthesizing images, for example, the sample "A boat sailing leisurely along the Seine River with the Eiffel Tower in background, Van Gogh style", resulting in higher quality videos.
>
> > **3. In User study, the 20 volunteers with expertise in image and video processing participated. Does the expert bias exist? that is to say, the participants are more tolerant to the defects of the methods? Will the ordinary participants from a consumer's perspective be better?**
>
> Thank you for the reviewers' suggestions. Following the advice of the reviewer, we have further invited seven undergraduate students to participate in our user study, all of whom are from non-computer science majors. In line with the settings of the user study in the main paper, we summarize the additional statistics as follows. Our method still outperforms other methods, and compared to the results in the main paper, the additional statistics slightly favor Vid.-Cond. Consistency.
>
> | Method | Video Quality | Vid.-Cond. Consistency | Overall score |
> | :--- | :---: | :---: | :---: |
> | AnimateDiff | 7.62% | 11.43% | 7.62% |
> | AnimateDiff + FreeInit | 33.33% | 12.38% | 21.90% |
> | AnimateDiff + I4VGen | **59.05%** | **76.19%** | **70.48%** |
>
> > **4. It's a trend that video generation tends to produce more diverse content with long duration, for example, the large camera pose change. Will the proposed method still work?**
>
> We provide more visual results involving significant changes in camera poses in **Fig. 16 of the supplementary material**, which demonstrate that our method can handle this scenario, improving the temporal consistency and smoothness of the synthesized videos.

---

> > ### Comment · Reviewer_tiD5 · 2024-11-26
> >
> > After reviewing the rebuttal, I am inclined to keep my rating.

---

> > > ### Author Response · Authors · 2024-11-27
> > > **Thanks for response**
> > >
> > > Thank you for your valuable suggestions and feedback, and we would like to express our gratitude for your time and insight.
> > >
> > > Best regards, The authors of Paper #7261

---

### Official Review · Reviewer_hdjr · 2024-11-02

**Soundness:** 3
**Presentation:** 2
**Contribution:** 2
**Rating:** 5
**Confidence:** 5

**Summary:**

This paper presents I4VGEN, a novel video diffusion inference pipeline that enhances pre-trained text-to-video models without additional training. It tackles the complexities of spatio-temporal modeling by utilizing advanced image techniques in two stages: first, synthesizing anchor images with a strategy to ensure visual realism and semantic accuracy; second, augmenting these images to generate videos through a noise-invariant video score distillation sampling (NI-VSDS) method. This process also includes a video regeneration step for refinement. Experiments show that I4VGEN significantly improves the visual quality and textual accuracy of generated videos and can be easily integrated into existing models, boosting overall video quality.

**Strengths:**

1. This paper introduces a training-free pipeline called I4VGen to improve the performance of text-to-video diffusion models throught image reference information.
2. A simple yet effective generation-selection strategy is proposed to obtain high-quality-images, while a noise-invariant video score distillation sampling is introduced for image animation.
3. Extensive experiments show that the proposed method comsiderably outperforms the performance of video diffusion baselines in terms of video quliaty.

**Weaknesses:**

1. The technical contributions of the paper are somewhat limited. The proposed noise-invariant video score distillation only modifies some hyper-parameters of the original SDS techinque.
2. Compared to the baseline results, the video actions enhanced using the proposed method in this paper are minimal or essentially stationary. The metrics in Table 1 also show that the proposed method heavily harm the dynamic degree of generated videos.
3. AnimateDiff relies on high-quality LoRAs to improve the quality and consistency of generated videos. Please provide generated videos of AnimateDiff with high-quality LoRAs for a fair comparison.

**Questions:**

1. Concerns about inference time. The proposed method consists of two stages: anchor image synthesis and anchor image-augmented video synthesis. The reviewer wants to know whether the time in the table includes the time used in the first stage.
2. Low-quality face video in Fig.6. In 1st row of Fig6, SparseCtrl produces a video in extremely low quality, could the authors explain reasons?

---

> ### Author Response · Authors · 2024-11-23
> **Thank you for your valuable feedbacks.**
>
> We would like to thank Reviewer for the constructive review. We did our best within the time constraints to address all of the points that you have raised, and will do our best to answer other concerns.
>
> > **1. The technical contributions of the paper are somewhat limited. The proposed noise-invariant video score distillation only modifies some hyper-parameters of the original SDS techinque.**
>
> NI-VSDS draws inspiration from SDS and represents a significant advancement within the realm of video diffusion models. As mentioned in Section 3.2 (Line 298), SDS is primarily employed in text-to-3D generation tasks, necessitating numerous optimization iterations for convergence. This protracted optimization process results in extended inference durations, rendering it unsuitable for text-to-video synthesis.
>
> In contrast, NI-VSDS is cleverly tailored for the video diffusion model. Unlike SDS, which optimizes from a randomly initialized representation, NI-VSDS optimizes from static videos and introduces a carefully designed optimization process. With a affordable computational burden (less than 50 optimization iterations), it achieves remarkable efficiency while retaining the ability to extract motion knowledge. These are all creative and essential for video synthesis tasks.
>
> > **2. Compared to the baseline results, the video actions enhanced using the proposed method in this paper are minimal or essentially stationary. The metrics in Table 1 also show that the proposed method heavily harm the dynamic degree of generated videos.**
>
>
> As we all know, dynamics degree and temporal consistency are mutually exclusive to a certain extent. Temporally inconsistent changes in the generated video will correspond to higher dynamics, which is undesirable. Therefore, higher motion dynamics are not necessarily optimal, and I4VGen strives for more reasonable motion dynamics.
>
> As shown in Fig. 1, 4, 8, both AnimateDiff and LaVie face a degree of temporal inconsistency. For example, in the case of "A person is painting in the room, Van Gogh style," there are dramatic and unreasonable changes in the scene development. Similar examples result in a high value of dynamic degree in Table 1. Conversely, our method has corrected this temporal inconsistency, as evidenced by the decrease in dynamic degree. In Table 3, we also provide results from user studies, which indicate that the introduction of I4VGen significantly improves the quality of the synthesized videos, aligning with user preferences.
>
> In fact, how to better handle the trade-off between dynamic degree and temporal consistency also reflects, to some extent, the performance of the video diffusion model. We know that LaVie's video generation performance is superior to AnimateDiff, so it's worth noting that after introducing I4VGen to correct temporal inconsistency, the corresponding dynamic degree did not decrease significantly.
>
> In addition, we also provide further experiments on motion instructions in Sec. D.2 of the supplementary material. By providing static instructions in negative prompt, the level of motion in the videos synthesized by I4VGen can be further enhanced while ensuring high quality. By offering specific motion instructions in prompt, I4VGen can accurately render this action.
>
>
> > **3. AnimateDiff relies on high-quality LoRAs to improve the quality and consistency of generated videos. Please provide generated videos of AnimateDiff with high-quality LoRAs for a fair comparison.**
>
> We provide the visual results on AnimateDiff using the Realistic Vision V5.1 LoRA in the **Fig. 14 of the supplementary material**. Our method still significantly improves the quality of the generated videos while showing excellent temporal consistency.
>
> In addition, we would like to emphasize that AnimateDiff is trained on the real video dataset, i.e., WebVid-10M. However, as the reviewer pointed out, AnimateDiff is dependent on high-quality LoRA for video synthesis. I4VGen can fully unleash the video generation capabilities of AnimateDiff, without relying on external LoRA.

---

> ### Author Response · Authors · 2024-11-23
> **Response (2)**
>
> > **4. Concerns about inference time. The proposed method consists of two stages: anchor image synthesis and anchor image-augmented video synthesis. The reviewer wants to know whether the time in the table includes the time used in the first stage.**
>
> The inference times provided in Table 4 include two stages: anchor image synthesis and anchor image-augmented video synthesis.
>
> > **5. Low-quality face video in Fig.6. In 1st row of Fig6, SparseCtrl produces a video in extremely low quality, could the authors explain reasons?**
>
> The first row of examples in Fig. 6 are out-of-domain for SparseCtrl and are difficult to handle. We also provide a visual result of DynamiCrafter for this example in **Fig. 11 of the supplementary material**. DynamiCrafter is a stronger image-to-video diffusion model than SparseCtrl, which exhibits state-of-the-art performance on the VBench Leaderboard. The video quality produced by DynamiCrafter is far superior to SparseCtrl.
>
> Furthermore, by introducing I4VGen, both SparseCtrl and DynamiCrafter are able to synthesize higher quality videos. This is due to the fact that I4VGen provides more condition image information to the video baselines, injecting the condition image into the noise space, which enhances their ability to utilize condition images.

---

> ### Author Response · Authors · 2024-11-27
> **Looking forward to the post-rebuttal feedback!**
>
> Dear Reviewer hdjr,
>
> We fully understand that this is a very busy period and appreciate your time and thoughtful suggestions and comments!
>
> As the discussion deadline is approaching, we are more than happy to provide any additional clarifications you might need.
>
> Best regards, The authors of Paper #7261

---

> ### Author Response · Authors · 2024-12-03
> **Official Comment by Authors**
>
> Dear Reviewer hdjr,
>
> Today is the last day of the discussion phase, and I would like to ask if our rebuttal has addressed your concerns.
>
> If our response has resolved your concerns, could you possibly increase your rating?
>
> Best regards,
>
> The authors of Paper #7261

---

> > ### Comment · Reviewer_hdjr · 2024-12-03
> >
> > Thank you for your detailed explanation. I acknowledge that NI-VSDS addresses certain limitations of SDS and provides optimizations tailored for video synthesis. However, while these adjustments improve efficiency, I still find the technical contributions to be incremental rather than fundamentally novel. For this reason, I am unable to raise my score.

---

### Official Review · Reviewer_dmyx · 2024-11-02

**Soundness:** 2
**Presentation:** 3
**Contribution:** 2
**Rating:** 6
**Confidence:** 4

**Summary:**

This paper proposes a video diffusion inference pipeline that leverages image generation techniques to enhance a pre-trained text-to-video (T2V) diffusion model. Instead of directly generating videos from noise, the method first utilizes a text-to-image (T2I) model to generate a high-quality anchor image. This image is then used to produce an initial video via Score Distillation Sampling (SDS) through the T2V model. A regeneration process is adopted to refine it, resulting in the final video. Experiments have been conducted to evaluate the effectiveness of the proposed method both qualitatively and quantitatively.

**Strengths:**

1. The proposed method uses a pre-trained image generation model to improve frame quality in text-to-video generation, which is helpful for high-quality video generation.
2. The presented results demonstrate good quality.

**Weaknesses:**

1. The proposed method appears to integrate the T2I model with SDS distillation for video generation, and the contribution seems incremental.
2. The motion observed in Fig. 1 appears to be smaller compared to the baselines.
3. There is a lack of analysis for different regeneration steps.

**Questions:**

1. What are the effects of using different image generation models? Can better video quality be achieved if a better image model (e.g., FLUX) is used to generate anchor images?
2. Will the SDS degrade image quality? It would be better to provide quantitative results in Table 2 to show its effects.
3. In Fig. 8, the prompt for the last video is mistaken.

---

> ### Author Response · Authors · 2024-11-23
> **Thank you for your valuable feedbacks.**
>
> We would like to thank Reviewer for the constructive review. We did our best within the time constraints to address all of the points that you have raised, and will do our best to answer other concerns.
>
> > **1. The proposed method appears to integrate the T2I model with SDS distillation for video generation, and the contribution seems incremental.**
>
> I will address your concerns from the following two perspectives:
>
> **(1) On T2I diffusion model**
>
> The contribution of this paper is a novel video diffusion inference pipeline, called I4VGen, which enhances the pre-trained text-to-video diffusion model by incorporating image reference information into the inference process, without requiring additional training or learnable parameters. In our experiments, we use the T2I diffusion model to obtain anchor images and achieve good performance.
>
> At the same time, as mentioned in sec. 4.5, our method also adapts to user-provided images (L451). As shown in Fig. 7 of the main paper, we use the real images as anchor images to produce high-fidelity videos.
>
> **(2) On NI-VSDS**
>
> NI-VSDS draws inspiration from SDS and represents a significant advancement within the realm of video diffusion models. As mentioned in Section 3.2 (Line 298), SDS is primarily employed in text-to-3D generation tasks, necessitating numerous optimization iterations for convergence. This protracted optimization process results in extended inference durations, rendering it unsuitable for text-to-video synthesis.
>
> In contrast, NI-VSDS has adeptly tailored video diffusion models, achieving remarkable **efficiency** with a affordable computational burden (less than 50 optimization iterations), while preserving the capacity to distill motion knowledge.
>
> > **2. The motion observed in Fig. 1 appears to be smaller compared to the baselines.**
>
> As is well known, dynamic degree and temporal consistency are somewhat mutually exclusive to some extent. Temporally inconsistent varies in the generated video will correspond to higher dynamism, but this is something we do not want to see. Therefore, higher motion dynamics are not necessarily optimal, I4VGen is dedicated to more reasonable motion dynamics.
>
> As shown in Fig. 1, I4VGen significantly improves the temporal consistency (top-left and bottom-right), visual realism (top-right), and semantic fidelity (bottom-left) of the synthesized videos. The video baseline achieves relatively high motion dynamics but introduces unwilling motion patterns, severely compromising temporal consistency, which is unacceptable. Reviewers can examine the corresponding videos in .mp4 format in the supplementary material. I4VGen significantly enhances temporal consistency while ensuring reasonable and accurate motion dynamics, without introducing additional motion information.
>
> In addition, we also provide further experiments on motion instructions in Sec. D.2 of the supplementary material. By providing static instructions in negative prompt, the level of motion in the videos synthesized by I4VGen can be further enhanced while ensuring high quality. By offering specific motion instructions in prompt, I4VGen can accurately render this action.

---

> ### Author Response · Authors · 2024-11-23
> **Response (2)**
>
> > **3. There is a lack of analysis for different regeneration steps.**
>
> We have included visual experiments on hyperparameter settings in Section B.2 of the supplementary material of the initial manuscript. I4VGen is a training-free method that enhances video generation performance by refining the inference process. It is evident that I4VGen is also a case-specific method, where different cases correspond to different optimal hyperparameters. In this paper, we present a relatively moderate empirical setting that is suitable for most instances.
>
> In fact, it is clear that as $p_{re}$​ increases, the utilization of anchor image information decreases. Both the method design and experimental experience indicate that for cases and methods demonstrating poor performance in temporal consistency, it is necessary to appropriately decrease $p_{re}$. We also provide a visualization of the impact of $p_{re}$ in the **Fig. 17 in the supplementary material**.
>
>
> > **4. What are the effects of using different image generation models? Can better video quality be achieved if a better image model (e.g., FLUX) is used to generate anchor images?**
>
> As mentioned in sec. 4.5, our method also adapts to user-provided images (L451). As shown in Fig. 7 of the main paper, we use real images as anchor images to produce high-fidelity videos.
>
> Regarding the impact of image models, as discussed in Section 4.4 of the main paper, in this study, we introduce a simple yet effective image generation-selection strategy to enhance the performance of the image model, thereby improving the performance of the video diffusion model. This suggests that higher-quality anchor images lead to higher-quality videos.
>
> We also acknowledge that simply improving the quality of anchor images does not necessarily lead to an improvement in video quality. This also depends on the distribution that the video baseline can handle. When the given anchor image exceeds what the baseline can manage, even more refined anchor images cannot further optimize the synthesized videos.
>
> We also provide the visual results of I4VGen adapted on FLUX in the **Fig. 13 in the supplementary material**. Despite the detailed and realistic images synthesized by FLUX, AnimateDiff + I4VGen is still constrained by the video baseline, i.e., AnimateDiff, in rendering image details and is unable to synthesize realistic videos. Evidently, the distribution of images synthesized by FLUX exceeds what AnimateDiff can handle, which relies on SD 1.5. However, the layout and composition information of images synthesized by FLUX still provide strong support for video synthesis, resulting in promising outcomes.
>
>
> > **5. Will the SDS degrade image quality? It would be better to provide quantitative results in Table 2 to show its effects.**
>
> Fig. 5 of the main paper and **Fig. 15 of the supplementary material** visualize intermediate results.  It is obvious that the dynamic images generated by NI-VSDS suffer from low single-frame quality to some extent, especially blurring in details. This blurring issue is also an inherent problem of SDS-like methods. Therefore, video regeneration is crucial to refine the appearance details.

---

> ### Author Response · Authors · 2024-11-27
> **Looking forward to the post-rebuttal feedback!**
>
> Dear Reviewer dmyx,
>
> We fully understand that this is a very busy period and appreciate your time and thoughtful suggestions and comments!
>
> As the discussion deadline is approaching, we are more than happy to provide any additional clarifications you might need.
>
> Best regards, The authors of Paper #7261

---

> ### Author Response · Authors · 2024-12-03
> **Official Comment by Authors**
>
> Dear Reviewer dmyx,
>
> Today is the last day of the discussion phase, and I would like to ask if our rebuttal has addressed your concerns.
>
> If our response has resolved your concerns, could you possibly increase your rating?
>
> Best regards,
>
> The authors of Paper #7261

---

> > ### Comment · Reviewer_dmyx · 2024-12-03
> > **Official Comment by Reviewer dmyx**
> >
> > Thank you to the authors for their efforts. I have read the rebuttal and the comments from other reviewers. Some of my concerns have been addressed, but I still think the novelty of this paper is limited. Although the authors state that the NI-VSDS has adeptly tailored video diffusion models, achieving remarkable efficiency, the difference between it and SDS is not significant (as also mentioned by Reviewer hdjr). Additionally, the distribution of the synthesized video is still limited by the capabilities of the T2V model, which diminishes the gains from the T2I model.
> >
> > Overall, applying the T2I model to improve T2V is interesting, and I will raise my rating to 6.

---

### Official Review · Reviewer_vp1P · 2024-11-04

**Soundness:** 2
**Presentation:** 3
**Contribution:** 2
**Rating:** 5
**Confidence:** 4

**Summary:**

This paper presented a two-stage text-to-Video Generation method I4VGEN, i.e., anchor image synthesis and anchor image-augmented text-to-video synthesis. Experiments are provided to assess its effectiveness in text2video generation and enhancing I2V methods.

**Strengths:**

+ Two-stage text-to-Video Generation method I4VGEN.
+ Can be integrated into existing image-to-video diffusion models.

**Weaknesses:**

- The integration with existing image-to-video diffusion models is interesting, but the authors are suggested to combined with more I2V models, especially several recent ones.
- More ablation studies are required to show whether the anchor image selection and the NI-VSDS  are optimal.
- In the bottom of Fig. 6, albeit better image quality, it seems that the motion of the proposed method is smaller than that by SparseCtrl. More experiments are suggested to assess this aspect.

**Questions:**

- The integration with existing image-to-video diffusion models is interesting, but the authors are suggested to combined with more I2V models, especially several recent ones.
- More ablation studies are required to show whether the anchor image selection and the NI-VSDS  are optimal.
- In the bottom of Fig. 6, albeit better image quality, it seems that the motion of the proposed method is smaller than that by SparseCtrl. More experiments are suggested to assess this aspect.

---

> ### Author Response · Authors · 2024-11-23
> **Thank you for your valuable feedbacks.**
>
> We would like to thank Reviewer for the constructive review. We did our best within the time constraints to address all of the points that you have raised, and will do our best to answer other concerns.
>
> > **1. The integration with existing image-to-video diffusion models is interesting, but the authors are suggested to combined with more I2V models, especially several recent ones.**
>
> We appreciate the reviewer's acknowledgement of the extension of the proposed method on the image-to-video diffusion model. Following the reviewer's suggestion, we have integrated I4VGen into DynamiCrafter [1], which exhibits state-of-the-art performance on the VBench Leaderboard [2]. Visual results are provided in **Fig. 11 of the supplementary material** (marked in red font to distinguish it from the initial manuscript). As shown in Fig. 11, the beginnings and endings of videos generated by DynamiCrafter suffer from low quality. For example, the front part of the face video generated by DynamiCrafter exhibits serious artifacts, and the face in the latter part are deformed. Our method alleviates these issues, which demonstrates that I4VGen can significantly improve the quality of videos synthesized by DynamiCrafter.
>
> [1] DynamiCrafter: Animating Open-domain Images with Video Diffusion Priors, ECCV 2024.\
> [2] VBench: Comprehensive Benchmark Suite for Video Generative Models, CVPR 2024.
>
> > **2. More ablation studies are required to show whether the anchor image selection and the NI-VSDS are optimal.**
>
> Thanks for the reviewer's suggestion. We address the reviewer's concerns from the following two points.
>
> **(1) On anchor image selection.**
>
> We provide the ablation study on anchor image selection strategy in Sec. 4.4, Table 2 and Fig. 5 of the main paper. With the aid of the reward model, the generation-selection strategy is a direct but effective method to obtain high-quality images corresponding to the provided text prompts, based on the given text-to-image diffusion model. As shown in the orange highlighted part of Table 2, compared to randomly synthesizing an anchor image, this strategy provides a higher quality anchor image, significantly improving the single frame quality of the generated video and the consistency with the text prompt (Line 421 of the main paper). Fig. 5 also verifies this point.
>
> In our experiments, $N$ is set to 16, meaning we pick the highest quality image as the anchor image from 16 randomly generated candidate images. More candidate image generation will bring more inference cost, and we consider that 16 is a moderate choice in our experiments, although it is foreseeable that more candidate image synthesis helps to obtain higher quality anchor images, leading to better video generation performance. In addition, it is also worth noting that, even when $N$ is set to 1, i.e., without using the generation-selection strategy (I4VGen (w/o gen.-sel.) in Table 2), the proposed method can still significantly improve the performance of the video baseline (AnimateDiff in Table 2). This insight comes from the fact that text-to-video generation is still inferior to image generation in terms of quality and diversity.
>
> **(2) On NI-VSDS**
>
> We provide the ablation study on NI-VSDS in Sec. 4.4 and Table 2 of the main paper. As shown in the yellow highlighted part of Table 2, NI-VSDS is crucial for ensuring the motion dynamics of the synthesized video (Line 428 of the main paper). Our method achieves an effective balance between motion intensity and overall video quality.
>
> Regarding the setting of hyperparameters, as mentioned in Sec. B.2 of the supplementary material. I4VGen is a training-free method that improves video generation performance by correcting the inference process. It is obvious that I4VGen is also a case-wise method, where different cases correspond to different optimal hyperparameters. In this paper, we provide a relatively moderate empirical setting that is suitable for most instances, which serves as a lower bound for the performance of I4VGen, significantly better than the video baseline, and facilitates large-scale quantitative comparisons.
>
> We also provide a visualization of the impact of hyperparameters in Fig. 9 of the supplementary material, which shows that carefully adjusted hyperparameters can achieve higher quality videos. I4VGen supports fine-tuning parameters according to each case to obtain higher quality videos.
>
> We appreciate the reviewers' suggestion for ablation studies on optimal settings. We believe that customizing hyperparameters for the best quantitative performance on the VBench benchmark is not necessary, as they don't exhibit universality for wild samples. Our goal is to achieve good performance without the need for excessive effort in manually designing hyperparameters.

---

> ### Author Response · Authors · 2024-11-23
> **Response (2)**
>
> > **3. In the bottom of Fig. 6, albeit better image quality, it seems that the motion of the proposed method is smaller than that by SparseCtrl. More experiments are suggested to assess this aspect.**
>
> As is well known, dynamic degree and temporal consistency are somewhat mutually exclusive to some extent. Temporally inconsistent varies in the generated video will correspond to higher dynamism, but this is undesirable. As shown in Fig. 6 of the main paper, SparseCtrl cannot synthesize reasonable results, despite having more severe motion changes. By introducing I4VGen, SparseCtrl's ability has been greatly improved.
>
> Importantly, I4VGen maximizes the depiction of motion instructions provided in the text prompts, such as "talking" and "standing" in Fig. 6. For additional motion information, such as "the panda wildly waving its hands," it is not overly shown in the video synthesized by I4VGen, as compared to the one synthesized by SparseCtrl. We believe this is reasonable.
>
> Furthermore, we also conduct experiments on action instructions. As shown in **Fig. 12 of the supplementary material**, we explore two prompt-based motion enhancement strategies: (a) By providing static descriptions in negative prompt, the dynamic intensity of the synthesized videos can be further enhanced. (b) By providing specific action instruction in the prompt, such as "waving its hands", the synthesized video accurately renders this action. These findings indicate that I4VGen does not compromise the dynamic nature of the synthesized videos but rather depicts more reasonable and accurate motion.

---

> ### Author Response · Authors · 2024-11-27
> **Looking forward to the post-rebuttal feedback!**
>
> Dear Reviewer vp1P,
>
> We fully understand that this is a very busy period and appreciate your time and thoughtful suggestions and comments!
>
> As the discussion deadline is approaching, we are more than happy to provide any additional clarifications you might need.
>
> Best regards, The authors of Paper #7261

---

> ### Author Response · Authors · 2024-12-03
> **Official Comment by Authors**
>
> Dear Reviewer vp1P,
>
> Today is the last day of the discussion phase, and I would like to ask if our rebuttal has addressed your concerns.
>
> If our response has resolved your concerns, could you possibly increase your rating?
>
> Best regards,
>
> The authors of Paper #7261

---

### Author Response · Authors · 2024-12-04
**Official Comment by Authors**

Thank you to AC for supporting the discussion, and thanks to the reviewers for their time and thoughtful comments and feedback!

Best regards,

The authors of Paper #7261

---

### Meta-Review · Area_Chair_yGvs · 2024-12-20

**Metareview:**

While the proposed I4VGen method offers practical improvements for text-to-video generation, concerns about its novelty (dmyx), incremental technical contributions (hdjr) and practicality (tiD5) remain. Despite efforts in the rebuttal to address these issues, some reviewers feel that the contributions, particularly NI-VSDS, are minor modifications of SDS. In addition, the trade-off between dynamic motion and temporal consistency raises questions about the method's general applicability. Given the borderline scores and unresolved concerns, the decision is to reject. The authors are encouraged to refine the contributions and address the reviewers’ feedback for future submissions.

**Additional Comments On Reviewer Discussion:**

Reviewers raised concerns about NI-VSDS's novelty, dynamic motion reduction, and inference cost. The authors argued NI-VSDS improves efficiency and provided ablations on image selection and hyperparameters. They clarified dynamic motion trade-offs and added comparisons with more models. Reviewers hdjr and dmyx (despite the slightly positive rating) maintained concerns about incremental novelty. The novelty concerns remain substantial although. The final decision reflects these balanced considerations.

---

### Decision · Program_Chairs · 2025-01-22

Reject